# Increased expression of SPRR1A is associated with a poor prognosis in pancreatic ductal adenocarcinoma

Kohei Yamakawa[1,2,3], Michiyo Koyanagi-Aoi[1,2,4], Keiichiro Uehara[1,2,5], Atsuhiro Masuda[3], Hiroaki Yanagimoto[6], Hirochika Toyama[6], Takumi Fukumoto[6], Yuzo Kodama[3], Takashi Aoi[1,2,4] *

1 Division of Advanced Medical Science, Graduate School of Science, Technology and Innovation, Kobe University, Kobe, Hyogo, Japan, 2 Department of iPS Cell Applications, Graduate School of Medicine, Kobe University, Kobe, Hyogo, Japan, 3 Division of Gastroenterology, Department of Internal Medicine, Kobe University Graduate School of Medicine, Kobe, Hyogo, Japan, 4 Center for Human Resource Development for Regenerative Medicine, Kobe University Hospital, Kobe, Hyogo, Japan, 5 Department of Diagnostic Pathology, Kobe University Graduate School of Medicine, Kobe, Hyogo, Japan, 6 Division of Hepato-Biliary-Pancreatic Surgery, Department of Surgery, Kobe University Graduate School of Medicine, Kobe, Hyogo, Japan

* takaaoi@med.kobe-u.ac.jp

## Abstract

### Objectives

Small proline-rich protein 1A (SPRR1A) is recognized as a squamous differentiation marker but is also upregulated in some non-squamous cancers. However, its expression in pancreatic ductal adenocarcinoma (PDAC) has not been investigated. This study elucidated the expression of SPRR1A in PDAC and its effect on the prognosis and malignant behavior of PDAC.

### Methods

We examined the SPRR1A expression by immunohistochemistry in 86 surgical PDAC cases and revealed the relationship between its expression and the prognosis of the PDAC patients. Furthermore, we overexpressed SPRR1A in pancreatic cancer cell lines (PK-1 and Panc-1) and assessed the phenotype and gene expression changes *in vitro*.

### Results

Among the 84 cases, excluding 2 with squamous differentiation, 31 (36.9%) had a high SPRR1A expression. The overall survival (median 22.1 months vs. 33.6 months, p = 0.0357) and recurrence-free survival (median 10.7 months vs. 15.5 months, p = 0.0298) were significantly lower in the high-SPRR1A-expression group than in the low-SPRR1A-expression group. A multivariate analysis indicated that a high SPRR1A expression (HR 1.706, 95% CI 1.018 to 2.862, p = 0.0427) and residual tumor status (HR 2.687, 95% CI 1.487 to 4.855, p = 0.00106) were independent prognostic factors. The analysis of TCGA transcriptome data demonstrated that the high-SPRR1A-expression group had a

**Data Availability Statement:** The original datasets analyzed in the current study were downloaded from the public databases, the GDC Data Portal (dbGaP accession number: phs000178.v11.p8,

Project ID: TCGA-PAAD, URL: https://portal.gdc.
cancer.gov/projects/TCGA-PAAD) and GTEx Portal
(dbGaP accession number: phs000424.v8.p2,
URL: https://gtexportal.org/home/datasets). The
RNA sequencing data of our in vitro experiments
are registered and available in GEO (accession
number: GSE186935, URL: https://www.ncbi.nlm.
nih.gov/geo). All the data generated by the analysis
are included in this paper and its Supporting
Information files.

**Funding:** This work was supported by grants from
JSPS KAKENHI (18H02796; T.A., 20J12977; K.Y.),
Research Center Network for Realization of
Regenerative Medicine (16817073) from the Japan
Agency for Medical Research and Development,
AMED (T.A. and M.K-A.), Akira Sakagami Fund for
Research and Education, Kobe University Graduate
School of Medicine (T.A. and M.K-A.) and
Research Assistance Funds from Shinryokukai
General Incorporated Association (T.A.). The
funders had no role in study design, data collection
and analysis, decision to publish, or preparation of
the manuscript.

**Competing interests:** The authors have declared
that no competing interests exist.

significantly worse prognosis than the low-SPRR1A-expression group, which supported our data. SPRR1A overexpression in PK-1 and Panc-1 did not result in remarkable changes to *in vitro* phenotypes, such as the cell proliferation, chemo-resistance, EMT, migration or global gene expression.

## Conclusion

Increased expression of SPRR1A is associated with a poor prognosis in PDAC and may serve as a novel prognostic marker. However, our *in vitro* study suggests that the SPRR1A expression may be a consequence, not a cause, of the aggressive behavior of PDAC.

## Introduction

Pancreatic cancer is a lethal disease with the poorest prognosis, with a 5-year survival rate of approximately 6%-9% [1,2], in various cancers. The number of deaths caused by pancreatic cancer more than doubled from 1990 to 2017, with 466,000 deaths reported worldwide in 2020 [3,4]. Pancreatic cancer is characterized by intratumor heterogeneity and a highly desmoplastic and immunosuppressive tumor microenvironment, which leads to resistance to chemotherapy and thus a poor prognosis [5,6]. One reason why its poor prognosis has not improved is that its pathogenesis, even in pancreatic ductal adenocarcinoma (PDAC), the most common type of pancreatic cancer, is still not fully understood. Consequently, only a few effective molecular-targeted therapies are clinically available for PDAC [7,8], and treatment options remain limited. To improve the prognosis, it is essential to understand the pathogenesis of PDAC and to discover biomarkers and therapeutic target molecules.

The expression of molecules not expressed in the original lineage of cancers is known to generally lead to a poor prognosis [9–11]. The *small proline-rich protein (SPRR) 1A* gene is a structural protein of the cornified envelope, which exerts a barrier function against the environment, in the epidermis [12] and is recognized as a marker for terminal squamous cell differentiation [13,14]. The expression of SPRR1A is not usually found in normal non-squamous tissues, and its increased expression has been reported in some types of non-squamous cell carcinoma (non-SCC), such as colorectal cancer and breast cancer [15]. However, the significance of SPRR1A expression in non-SCC is poorly understood.

The *SPRR* gene family consists of 10 members, including *SPRR1B*, six *SPRR2*, one *SPRR3*, and one *SPRR4*, as well as *SPRR1A*, and all *SPPR* genes function as specific cornified envelope precursors [15]. Previous studies have shown that SPRR3 promoted the proliferation of breast cancer and colorectal cancer cells via the AKT and mitogen-activated protein kinase (MAPK) pathways [16,17] and that SPRR2B facilitated the growth of gastric cancer via the MDM2-p53/p21 signaling pathway [18], suggesting that SPRR family genes may be involved in cancer growth signaling. Furthermore, recent studies have reported the prognostic value of a high expression of SPRR1A in colon cancer, breast cancer and diffuse large B-cell lymphoma [19–21]. We have also found that SPRR1A may be associated with the characteristics of cancer stem cells derived from osteosarcoma [22].

According to these findings, SPRR1A may involve the pathogenesis of non-SCC, leading to a worse prognosis for cancer patients. However, there is no research focusing on the biological features of SPRR1A in cancers nor the expression of SPRR1A in PDAC. Therefore, the present study elucidated the expression of SPRR1A in PDAC and its effect on the prognosis and pathogenesis.

## Materials and methods

### Patient population

Surgical specimens were acquired from all 86 patients with stage II or III PDAC who underwent pancreatectomy between March 2011 and January 2017 at Kobe University Hospital. Clinical information for each patient was obtained from chart review. All data were anonymized and are shown in S1 Table.

This study was approved by the Institutional Review Board (IRB) for Clinical Research at Kobe University Hospital (approval number: B200179) and performed according to the Declaration of Helsinki principles. The IRB allowed a waiver of prospective informed consent, and this study information was disclosed to the public on our hospital website, providing the eligible patients with an opportunity to opt out.

### Immunohistochemistry (IHC)

All specimens were acquired from the 86 total individuals with PDAC, excluding cases without formalin-fixed paraffin-embedded (FFPE) samples, as described above. Four-micron-thick FFPE human tissue sections were processed.

All wash steps were performed at room temperature for 3 min each unless otherwise noted. In brief, slides were deparaffinized in xylene (3 washes, 5 min in first wash only) and rehydrated in graded dilutions of aqueous ethanol (EtOH; 2 washes in 100% EtOH; 1 wash in 95% EtOH; 1 wash in 70% EtOH). Slides were washed once in double-distilled $H_2O$ ($ddH_2O$) before being placed in an antigen target retrieval solution, pH 8 EDTA, and pressure cooked (3 min) for antigen retrieval. Slides were allowed to cool to room temperature and washed 3 times with 1x phosphate-buffered saline (PBS) with Tween-20, and then the tissue was blocked for endogenous peroxidase activity for 10 min using 0.3% $H_2O_2$/methanol. Slides were washed 3 times with 1x PBS and then incubated for 10 min in Blocking One Histo (Nacalai Tesque). Slides were washed 3 times with 1x PBS, incubated overnight at 4°C with SPRR1A rabbit antibody (1:200; Abcam plc, Cambridge, UK; catalog number: ab125374) or normal rabbit IgG (FUJIFILM Wako Pure Chemical Corporation, Osaka, Japan; catalog number: 148–09551). The following morning, the slides were washed 3 times with 1x PBS and then incubated using a commercial histofine simple stain MAX-PO (MULTI) kit (Nichirei Biosciences Inc., Tokyo, Japan) for 30 min. Slides were washed 3 times in 1x PBS before incubation with a DAB Substrate kit (Nichirei Biosciences Inc.) for 1 min and then washed twice in $ddH_2O$ and counterstained using a commercial hematoxylin solution (Sakura Finetek Japan Co., Tokyo, Japan). Excess dye was removed using 3 washes in $ddH_2O$. Tissues were dehydrated in aqueous EtOH (1 wash in 70% EtOH; 1 wash in 95% EtOH; 2 washes in 100% EtOH) and incubated in xylene (3 washes) before being coverslipped. All stained slides were digitized using a fluorescence microscope (BZ-X700; Keyence, Osaka, Japan).

The immunostaining of CK5/6 was performed using CK5/6 mouse antibody (1:100; Agilent, Santa Clara, CA, USA; catalog number: M7237) and VENTANA BenchMark GX (Roche Diagnostics K.K., Tokyo, Japan) according to the manufacturer's instructions.

### Hematoxylin-eosin (HE) staining

Slides were deparaffinized in xylene and rehydrated in graded dilutions of aqueous EtOH as described above. Slides were washed once in $ddH_2O$ and stained using a commercial eosin and hematoxylin solution (Sakura Finetek Japan Co.). Excess dye was removed using one wash in $ddH_2O$. Tissues were dehydrated in aqueous EtOH and incubated in xylene before being coverslipped as described above.

## The evaluation of squamous differentiation and SPRR1A expression in PDAC cases

All slides from PDAC patients were reviewed by two experienced physicians—a gastroenterologist (K.Y.) and a pathologist (K.U.)—who were both unaware of the clinical information of each case. For cases with different diagnoses between these physicians, they reviewed the slides together and reconciled the diagnoses.

Squamous differentiation was determined using HE staining, corroborated by immunostaining of CK5/6 as a diagnostic aid.

The SPRR1A expression was evaluated using the staining intensity of the normal pancreatic ductal epithelium as a reference (S1A Fig). The normal esophageal epithelium was used as a positive control for SPRR1A staining, and isotype IgG was used as a negative control to optimize the antibody staining conditions (S1B Fig). PDAC specimens with a higher staining intensity of SPRR1A than the normal pancreatic ductal epithelium were defined as having a high SPRR1A expression, whereas specimens with a staining intensity of SPRR1A equal to or lower than that of the normal pancreatic ductal epithelium were defined as having a low SPRR1A expression.

The overall survival (OS) was defined as the time between surgery and death, and the recurrence-free survival (RFS) was defined as the time between surgery and disease recurrence.

## Status of four driver genes in PDAC cases

This study analyzed alterations in the *KRAS*, *TP53*, *CDKN2A/p16*, and *SMAD4* genes using next-generation sequencing (NGS), droplet digital PCR (ddPCR), and IHC as described previously [23]. For the *KRAS* mutation, tumors were classified as "Present" or "Absent" based on NGS. For *TP53* alterations, tumors were classified as "Present" or "Absent" using a combination of NGS, ddPCR, and IHC. For *CDKN2A/p16* and *SMAD4* alterations, tumors were classified as "Present" or "Absent" based on IHC.

## The Cancer Genome Atlas (TCGA) and Genotype-Tissue Expression (GTEx) data analyses

The TCGA-PAAD, a pancreatic cancer dataset, and GTEx data, a normal tissues dataset, were downloaded from the GDC Data Portal (https://portal.gdc.cancer.gov) and GTEx Portal (https://gtexportal.org/home/datasets), respectively.

For prognostic analyses, TCGA samples were stratified by the transcript level of *SPRR1A* into three groups: high (fragments per kilobase of exon per million reads mapped [FPKM] > 8.30, n = 59), moderate (FPKM 0.87 to 8.30, n = 59) and low (FPKM < 0.87, n = 59) (listed in S2 Table). The OS was estimated based on the Kaplan-Meier method and compared by log-rank test. In TCGA analyses, the OS was the time between the date of the diagnosis and death. A correlation analysis between *SPRR1A* and *KRT5* was performed using Pearson's product-moment correlation coefficient.

For comparing gene expression profiles between the high- and low-SPRR1A-expression groups, we identified differentially expressed entities using an unpaired *t*-test (p < 0.05). A gene ontology (GO) analysis of the identified entities was performed using g:Profiler (https://biit.cs.ut.ee/gprofiler/gost) [24] to extract significant GO terms (p < 0.001). K-means cluster analyses were carried out using Python 3.7.12 based on the expression of the signature genes of the molecular subtypes of PDAC.

For the analyses in Fig 1A, TCGA-PAAD data units were converted to transcripts per million (TPM) to compare TCGA-PAAD and GTEx data. An *SPRR1A* expression higher than the

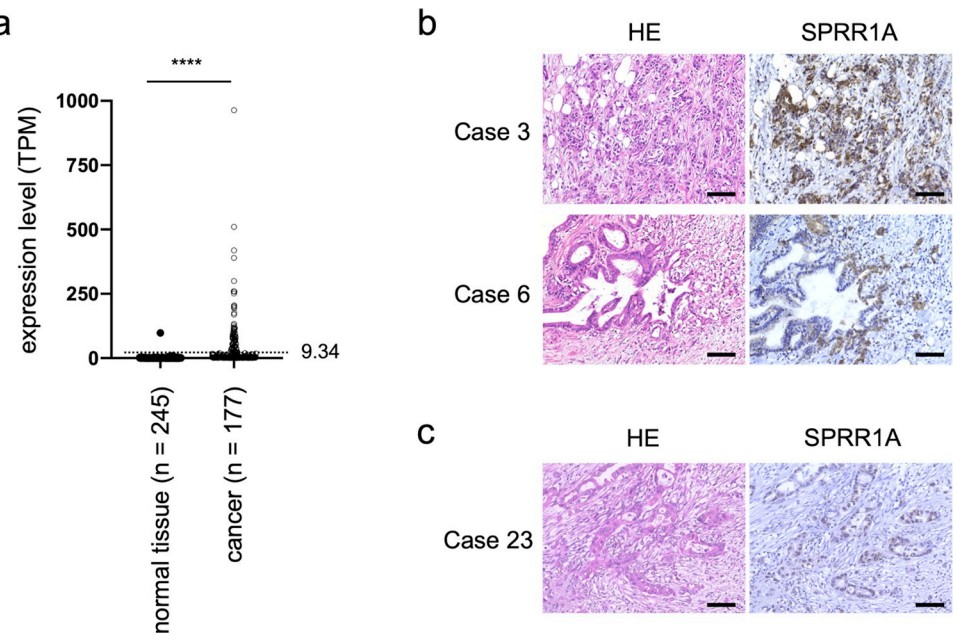

**Fig 1. The expression of SPRR1A in PDAC.** (a) The comparison of the SPRR1A expression (TPM) between normal pancreas tissue (n = 245) and pancreatic cancer (n = 177). The SPRR1A expression was elevated in 89 of 177 cases. ****p < 0.0001, unpaired *t*-test. The dotted line shows TPM 9.34, the second-highest expression in normal pancreas tissue. (b) Representative cases from the SPRR1A high expression group in PDAC (Cases 3 and 6). Scale bars, 500 μm. (c) Representative case from the SPRR1A low expression group in PDAC (Case 23). Scale bars, 500 μm.

second-highest expression in normal pancreas tissue (TPM 9.34) was considered elevated in cancer cases.

## Cell culture

We purchased the human pancreatic cancer cell lines (PK-1, PK-8, KLM-1, Panc-1 and MIA-Paca2) from RIKEN BioResouce Research Center (RIKEN BRC, Ibaraki, Japan) and another human pancreatic cancer cell line (BxPC-3) and Plat-A amphotropic retrovirus packaging cells from American Type Culture Collection (ATCC, Manassas, VA, USA). We maintained PK-1, PK-8, KLM-1, Panc-1 and BxPC-3 in RPMI-1640 (Nacalai Tesque) supplemented with 10% fetal bovine serum (FBS) (Sigma-Aldrich, St. Louis, MO, USA), 100 U/ml penicillin (Life Technologies, Carlsbad, CA, USA) and 100 μg/ml streptomycin (Life Technologies) at 37°C in a humidified 5% $CO_2$ incubator. We maintained MIAPaca2 and Plat-A in DMEM (Nacalai Tesque) supplemented with 10% FBS, 100 U/ml penicillin and 100 μg/ml streptomycin. In Plat-A culture, 1 μg/ml of puromycin (Nacalai Tesque) and 10 μg/ml of blasticidin (Funakoshi, Tokyo, Japan) were added.

## Retroviral infection and plasmid transfection

We designed the polycistronic retroviral vector (pMXs-*SPRR1A*) and plasmid vector (pCAG-*SPRR1A*) encoding *SPRR1A* and *FLAG*. In brief, human *SPRR1A* cDNA were amplified by nested polymerase chain reaction (PCR) with primers containing BamHI and HindIII site using normal esophagus tissues cDNA synthesized from total RNA (Biochain Institute Inc, Newark, CA, USA; catalog number: R1234106-50) as a template and cloned between the BamHI-HindIII site of the pCAG-*FLAG*(C) vector (pCAG-*SPRR1A*). To generate the pMXs-*SPRR1A-FLAG* vector (pMXs-*SPRR1A*), the *SPRR1A-FLAG* construct was extracted from

pCAG-*SPRR1A* using the restriction enzyme of the BamHI and XhoI site and cloned into the BamHI-XhoI site of pMXs vector.

One day before transfection, Plat-A packaging cells were seeded at $1.5 \times 10^6$ cells per 10-cm dish. The next day, the cells were transfected with 9 μg of pMXs-*SPRR1A* using the Fugene HD transfection reagent (Promega, Madison, WI, USA) according to the manufacturer's instructions. Concomitantly, pMXs with *DsRed* (pMXs-*DsRed*) (Addgene, Watertown, MA, USA; catalog number: 22724) was used as a control vector. Twenty-four hours after transfection, the Plat-A medium was replaced, and the pancreatic cancer cell lines (PK-1 and Panc-1) were seeded at $5 \times 10^5$ cells per 60-mm dish. After 24 h, virus-containing supernatants derived from these Plat-A cultures were filtered through a 0.45-μm cellulose acetate filter (Cytiva, Tokyo, Japan), supplemented with 4 μg/ml polybrene (Nacalai Tesque), and added to target cells immediately. Cell growth of the infected cells was evaluated by cell count using the Countess system II (Life Technologies) for each passage.

One day before transfection, PK-1 was seeded at $1.2 \times 10^5$ cells per 6-well plate. The next day, the cells were transfected with 9 μg of pCAG-*SPRR1A* vectors using the Fugene HD transfection reagent (Promega) according to the manufacturer's instructions. Concomitantly, pCAG with *GFP* (pCAG-*GFP*) was used as a control vector. Forty-eight hours after transfection, total RNA was isolated as described below.

## Semi-quantitative or real-time quantitative reverse transcription-PCR (RT-PCR)

Total RNA was isolated using TRIzol (Life Technologies) and treated with the TURBO DNA-free kit (Life Technologies). The Prime Script II 1st strand cDNA Synthesis Kit (Takara, Shiga, Japan) synthesized cDNA from 500 ng of total RNA. For semi-quantitative RT-PCR, the resulting cDNA was subjected to PCR with a Takara Ex Taq PCR kit (Takara). PCR conditions were as follows: 94˚C for 1 min, followed by 25 or 30 cycles of denaturing at 94˚C for 10 s, annealing at 60˚C for 15 s and extension at 72˚C for 15 s. Real-time quantitative RT-PCR analyses (Fig 4D) were performed using TB Green Premix Ex Taq (Takara) on a Light Cycler 480 II (Roche, Basel, Switzerland) according to the manufacturer's instructions. The PCR primers are listed in S3 Table.

## Cell proliferation assays

A total of $2 \times 10^3$ cells was seeded in 96-well plates. The number of viable cells was assessed at days 0, 1 and 3 by measuring cellular ATP levels using CellTiter-Glo® (Promega) according to the manufacturer's instructions.

## Gemcitabine (Gem) chemo-resistance analyses

A total of $2 \times 10^3$ cells were seeded with RPMI containing DMSO or 10 nM Gem (FUJIFILM Wako Pure Chemical Corporation) in 96-well plates. After incubation for three days, the cell viability was assessed by measuring the cellular ATP levels using CellTiter-Glo® (Promega) according to the manufacturer's instructions. The cell viability in the presence of Gem was calculated as a percentage of the viability in its absence.

## Cell migration assays

The migration ability of cells was evaluated using a scratch wound healing assay. Cells were seeded in 12-well plates at a density of $5 \times 10^4$ cells/well and allowed to reach 80%-90% confluence. A wound was artificially created by scratching the cell monolayer with a 200 μl pipette

tip. Plates were washed with PBS to remove detached cells and maintained for 48 h. Wound closure was observed at 0, 12, 24, 36 and 48 h. The migration area (MA) in each group was calculated using the Image J software program (Java image processing program inspired by the National Institute of Health, USA) according to the following equation: MA = the area of the scratch at 0 h–the area of the scratch at 24 h. The MA value of the pMXs-DsRed population was used as a reference. The following equation determined the relative cell migration ability: Relative cell migration ability = MA (pMXs-SPRR1A) / MA (pMXs-DsRed).

## Western blotting

Aggregates were washed with PBS and lysed in M-PER lysis buffer (Thermo Fisher Scientific) supplemented with cOmplete Protease Inhibitor Cocktail (Sigma-Aldrich). A bicinchoninic acid (BCA) assay determined the protein concentration of cell lysates. Five micrograms of protein per lane were subjected to sodium dodecyl sulfate-polyacrylamide gel electrophoresis (SDS-PAGE) and then transferred to a PVDF membrane. After the membrane was incubated with primary antibody and HRP-conjugated secondary antibody, signals were visualized with Immobilon Western Chemiluminescent HRP Substrate (Merck KGaA, Darmstadt, Germany). Images were obtained using Amersham Imager 600 (Cytiva).

The following primary antibodies were used: a rabbit anti-human SPRR1A (1:500; Abcam, Cambridge, UK, catalog number: ab125374) and a mouse anti-human β-actin (1:3000; Sigma-Aldrich; catalog number: A5441). The following secondary antibodies were used: HRP-conjugated anti-mouse IgG (1:3000; Cell Signaling Technology, Danvers, MA, USA; catalog number: #7076) and HRP-conjugated anti-rabbit IgG (1:2500; Promega; catalog number: W4011).

## RNA sequencing

Total RNA was isolated and treated with DNase as described above. The RNA was sent to Macrogen (Seoul, South Korea; https://www.macrogen.com) for library preparation and paired-end RNA sequencing on the Illumina NovaSeq6000 platform. Raw sequence files (fastq) were aligned to the human transcriptome (hg38) reference sequences using the Strand NGS software program (Strand Life Science, Karnataka, India) with default parameters. The aligned reads were normalized using TPM. A gene ontology (GO) analysis of the obtained RNA sequencing data was performed using the Strand NGS software program (Strand Life Science).

RNA sequencing data were registered in the Gene Expression Omnibus (GEO) with accession number GSE186935.

## Statistical analyses

All of the results are shown as the mean plus standard deviation (s.d.). Statistical significance between groups of data was analyzed using the GraphPad Prism 8 software program (GraphPad Software, San Diego, CA, USA). A two-tailed Student's *t*-test and the chi-square test (or Fisher's exact test where appropriate) were used for the statistical comparison between the two groups. Pearson's correlation analysis was used to explore the correlation between SPRR1A and the signature genes of the molecular subtypes of PDAC. Kaplan–Meier estimates were compared using stratified log-rank tests. Univariate and multivariate analyses were performed with EZR (Saitama Medical Center, Jichi Medical University, Saitama, Japan) [25], which is a graphical user interface for R (The R Foundation for Statistical Computing, Vienna, Austria, version 4.0.4). Hazard ratios (HRs) and the corresponding 95% confidence intervals (CIs) were estimated using Cox proportional-hazards models. A p-value lower than 0.05 ($p < 0.05$) was considered statistically significant.

## Results

### Expression of SPRR1A in PDAC

To assess the expression of SPRR1A in pancreatic cancers, we initially compared the transcript levels of *SPRR1A* between normal pancreas tissues and pancreatic cancers by analyzing RNA sequencing data from two databases (TCGA and GTEx). The transcript levels of *SPRR1A* in pancreatic cancers were elevated in 89 of 177 cases and were significantly higher than in normal pancreatic tissues (mean TPM 1.28 vs. 45.99, $p < 0.0001$) (Fig 1A).

Next, we examined the expression of SPRR1A protein in PDAC by IHC staining using surgical specimens from 86 consecutive patients with stage II or III PDAC who underwent pancreatectomy between March 2011 and January 2017 at Kobe University Hospital. Two of the 86 PDAC specimens (Case 12 and 41) had squamous differentiation, which was determined using HE staining and immunostaining of CK5/6, a squamous epithelial marker [26], in some areas of the PDAC region (<30%). The regions with squamous differentiation had a high SPRR1A expression (S1C Fig). In normal squamous epithelial tissues and urothelial carcinoma with squamous differentiation, increased expression of SPRR1A has been reported as a result of terminal squamous cell differentiation [15,27]. To focus on the significance of SPRR1A expression in non-squamous cell carcinoma, the current study excluded these two specimens with squamous differentiation (Case 12 and 41) (S1D Fig). The remaining 84 cases included 15 with stage IIa, 55 with stage IIb and 14 with stage III. The mean age of the patients was 69 (40–85) years old, and the proportion of women was slightly higher than men (detailed in Table 1). IHC staining showed that the PDAC regions of 31 (36.9%) specimens, including Cases 3 and 6 (Fig 1B), were strongly stained for SPRR1A compared to the normal pancreatic ductal epithelium (S1A Fig). In contrast, 53 (63.1%) specimens, including Case 23 (Fig 1C), exhibited staining equal to or weaker than the normal pancreatic ductal epithelium; we classified the former as the high-SPRR1A-expression group and the latter as the low-SPRR1A-expression group and then used them for the subsequent analyses (detailed in *Materials and Methods*) (S1D Fig).

The high-SPRR1A-expression group showed the following two main expression patterns: in well and moderately differentiated carcinoma, SPRR1A was expressed mainly in invasive areas (Case 6), while in poorly differentiated carcinoma, SPRR1A was patchily expressed (Case 3) (Fig 1B).

### Increased expression of SPRR1A is associated with a worse prognosis in PDAC patients

To examine whether or not the expression of SPRR1A was associated with the prognosis in PDAC patients, we compared the prognoses between the high and low-SPRR1A-expression groups. The OS was significantly lower in the high-SPRR1A-expression group than in the low-SPRR1A-expression group (median OS 22.1 months vs. 33.6 months, $p = 0.0357$) (Figs 2A and S1A). The RFS was also significantly lower in the high-SPRR1A-expression group than in the low-SPRR1A-expression group (median RFS 10.7 months vs. 15.5 months, $p = 0.0298$) (Figs 2B and S2B). Due to the significant influence of the pathological stage and residual tumor status on the patient prognosis, we excluded stage III and R1 cases, respectively, and assessed the prognostic value of the *SPRR1A* expression again. In the analysis excluding stage III cases, the OS was significantly lower in the high-SPRR1A-expression group than in the low-SPRR1A-expression group (median OS 22.1 months vs. 33.7 months, $p = 0.0322$) (S2C Fig). In the analysis excluding R1 cases, the OS was significantly lower in the high-SPRR1A-expression group than in the low-SPRR1A-expression group (median OS 22.0 months vs. 37.0 months, $p = 0.0279$) (S2D Fig).

**Table 1. Difference in patient characteristics between the high and low-SPRR1A-expression groups.**

| | | All cases | | SPRR1A | | | | |
| --- | --- | --- | --- | --- | --- | --- | --- | --- |
| | | | | High expression | | Low expression | | |
| | | N = 84 | | N = 31 | | N = 53 | | *P* value |
| Age, years; median (range) | | | | | | | | 0.022 |
| | | 69 (40–85) | | 65 (51–79) | | 72 (40–85) | | |
| Gender | | | | | | | | 0.500 |
| | Female | 34 | (40.5%) | 11 | (35.5%) | 23 | (43.4%) | |
| | Male | 50 | (59.5%) | 20 | (64.5%) | 30 | (56.6%) | |
| BMI (kg/m$^2$), median ± s.d. | | | | | | | | 0.788 |
| | | 21.0 ± 3.52 | | 21.0 ± 2.92 | | 21.2 ± 3.81 | | |
| CEA (ng/ml), mean ± s.d. | | | | | | | | 0.851 |
| | | 6.22 ± 10.1 | | 5.95 ± 10.0 | | 6.38 ± 10.2 | | |
| CA19-9 (U/ml), mean ± s.d. | | | | | | | | 0.994 |
| | | 590.9 ± 1184.3 | | 589.7 ± 899.4 | | 591.6 ± 1322.8 | | |
| Pathological stage# | | | | | | | | 0.189 |
| | IIa | 15 | (17.9%) | 3 | (9.7%) | 12 | (22.6%) | |
| | IIb | 55 | (65.5%) | 24 | (77.4%) | 31 | (58.5%) | |
| | III | 14 | (16.7%) | 4 | (12.9%) | 10 | (18.9%) | |
| T factor | | | | | | | | 0.256 |
| | 1c | 2 | (2.4%) | 1 | (3.2%) | 1 | (1.9%) | |
| | 2 | 34 | (40.5%) | 16 | (51.6%) | 18 | (34.0%) | |
| | 3 | 45 | (53.6%) | 14 | (45.2%) | 31 | (58.5%) | |
| | 4 | 3 | (3.6%) | 0 | (0.0%) | 3 | (5.7%) | |
| N factor | | | | | | | | 0.169 |
| | 0 | 17 | (20.2%) | 3 | (9.7%) | 14 | (26.4%) | |
| | 1 | 56 | (66.7%) | 24 | (77.4%) | 32 | (60.4%) | |
| | 2 | 11 | (13.1%) | 4 | (12.9%) | 7 | (13.2%) | |
| Histological grade | | | | | | | | 0.363 |
| | Well-differentiated | 22 | (26.2%) | 9 | (29.0%) | 13 | (24.5%) | |
| | Moderately differentiated | 53 | (63.1%) | 17 | (54.8%) | 36 | (67.9%) | |
| | Poorly differentiated | 9 | (10.7%) | 5 | (16.1%) | 4 | (7.5%) | |
| Residual tumor status | | | | | | | | > 0.999 |
| | R0 | 65 | (77.4%) | 24 | (77.4%) | 41 | (77.4%) | |
| | R1 | 19 | (22.6%) | 7 | (22.6%) | 12 | (22.6%) | |
| | R2 | 0 | (0.0%) | 0 | (0.0%) | 0 | (0.0%) | |
| Peritoneal lavage cytology | | | | | | | | NA |
| | CY0 | 84 | (100.0%) | 31 | (100.0%) | 53 | (100.0%) | |
| | CY1 | 0 | (0.0%) | 0 | (0.0%) | 0 | (0.0%) | |
| Neoadjuvant chemotherapy | | | | | | | | 0.062 |
| | Absent | 71 | (84.5%) | 23 | (74.2%) | 48 | (90.6%) | |
| | Present | 13 | (15.5%) | 8 | (25.8%) | 5 | (9.4%) | |
| Adjuvant chemotherapy | | | | | | | | > 0.999 |
| | Absent | 26 | (31.0%) | 10 | (32.3%) | 16 | (30.2%) | |
| | Present | 58 | (69.0%) | 21 | (67.7%) | 37 | (69.8%) | |
| KRAS mutation | | | | | | | | 0.668 |
| | Absent | 5 | (6.0%) | 2 | (6.5%) | 3 | (5.7%) | |
| | Present | 79 | (94.0%) | 29 | (93.5%) | 50 | (94.3%) | |
| TP53 alteration | | | | | | | | 0.812 |

(*Continued*)

**Table 1.** (Continued)

| | | All cases | | SPRR1A | | | | |
| --- | --- | --- | --- | --- | --- | --- | --- | --- |
| | | | | High expression | | Low expression | | |
| | | N = 84 | | N = 31 | | N = 53 | | *P* value |
| | Absent | 26 | (31.0%) | 9 | (29.0%) | 17 | (32.1%) | |
| | Present | 58 | (69.0%) | 22 | (71.0%) | 36 | (67.9%) | |
| CDKN2A/p16 alteration | | | | | | | | 0.809 |
| | Absent | 27 | (32.1%) | 9 | (29.0%) | 18 | (34.0%) | |
| | Present | 57 | (67.9%) | 22 | (71.0%) | 35 | (66.0%) | |
| SMAD4 alteration | | | | | | | | 0.176 |
| | Absent | 49 | (58.3%) | 15 | (48.4%) | 34 | (64.2%) | |
| | Present | 35 | (41.7%) | 16 | (51.6%) | 19 | (35.8%) | |

[#]Pathological stage was classified according to the UICC 8th edition.

BMI, body mass index; CEA, carcinoembryonic antigen; CA19-9, carbohydrate antigen 19–9; NA, not available; s.d., standard deviation.

Next, we analyzed the relationship between the transcript level of *SPRR1A* and prognosis in PDAC using the TCGA-PAAD dataset to confirm our data. In TCGA analyses, we stratified 177 patients with pancreatic cancers by the transcript level of *SPRR1A* into three groups. The TCGA analysis clarified that the high and moderate *SPRR1A* groups had a significantly worse prognosis than the low *SPRR1A* group (median OS 50.1 months vs. 16.0 months vs. 17.7 months, p = 0.0020) (Figs 3A and S3A). Squamous differentiation, unlike the adenocarcinoma component, essentially expresses *SPRR1A*, as described above. After verifying that the expression of *keratin 5 (KRT5)* did not correlate with that of *SPRR1A* (S3B Fig), we used the same method as in S1C and S1D Fig to exclude the eight patients with high transcript levels of *KRT5*, a squamous epithelial marker [19], above the average plus two s.d. (S3C Fig). We analyzed the TCGA-PAAD dataset again to assess the prognostic value of *SPRR1A* expression in non-squamous PDAC and obtained a similar result (median OS 35.3 months vs. 16.0 months vs. 18.2 months, p = 0.0086) (Fig 3B).

## Differing characteristics between the high and low-SPRR1A-expression groups

To examine the relationship between the expression of SPRR1A and the clinical characteristics, we compared the clinical characteristics of patients between the high and low-SPRR1A-expression groups (Table 1). Aside from patients in the high-SPRR1A-expression group being significantly younger than those in the low-SPRR1A-expression group (median age 65 vs. 72 years old, p = 0.022), there was no significant difference in the following variables: gender, body mass index (BMI), carcinoembryonic antigen (CEA), CA19-9, pathological stage, T factor, N factor, histological grade, residual tumor status, peritoneal lavage cytology, neoadjuvant chemotherapy, adjuvant chemotherapy or four major driver mutations of PDAC (*KRAS*, *TP53*, *CDKN2A/p16* or *SMAD4*).

## High expression of SPRR1A in PDAC is an independent prognostic factor

To explore the relative contributions of each variable to the OS in PDAC patients, we performed univariate and multivariate analyses using a Cox proportional hazard model (Table 2). In the univariate analysis, significant risk factors were residual tumor status of R1 (HR 2.498, 95% CI 1.414 to 4.415, p = 0.00163) and high SPRR1A expression (HR 1.716, 95% CI 1.031 to

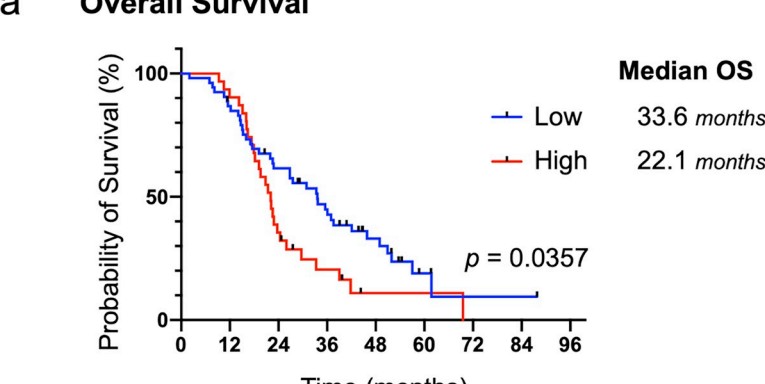

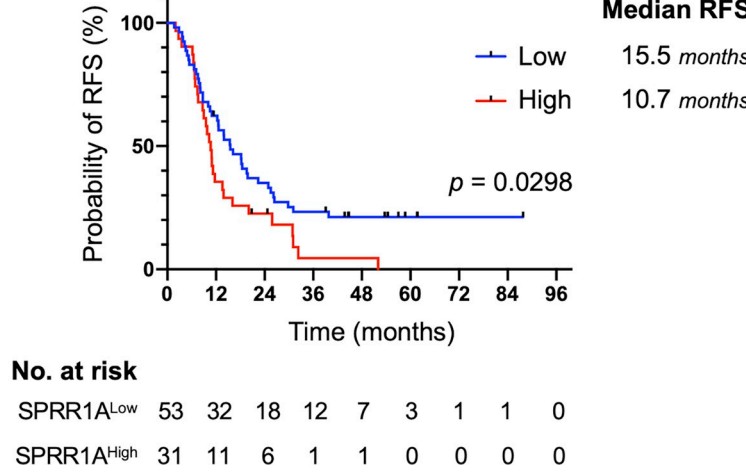

**Fig 2. An increased protein expression of SPRR1A was associated with a poor prognosis in PDAC patients.** (a) Kaplan-Meier estimates of the OS stratified by the SPRR1A expression. (b) Kaplan-Meier estimates of the RFS stratified by the SPRR1A expression. Tick marks indicate censored data. Kaplan–Meier estimates were compared using a stratified log-rank test.

2.856, p = 0.0378). In addition, we performed a multivariate analysis for the 4 variables with p < 0.20 in the univariate analysis, extracting a residual tumor status of R1 (HR 2.687, 95% CI 1.487 to 4.855, p = 0.00106) and high SPRR1A expression (HR 1.706, 95% CI 1.018 to 2.862, p = 0.0427) as significant risk factors.

## SPRR1A overexpression did not influence the phenotype in pancreatic cancer cells in vitro

To examine whether or not SPRR1A overexpression leads to aggressive behavior in PDAC, we initially used PK-1, a well-differentiated PDAC, which is the most common pancreatic cancer,

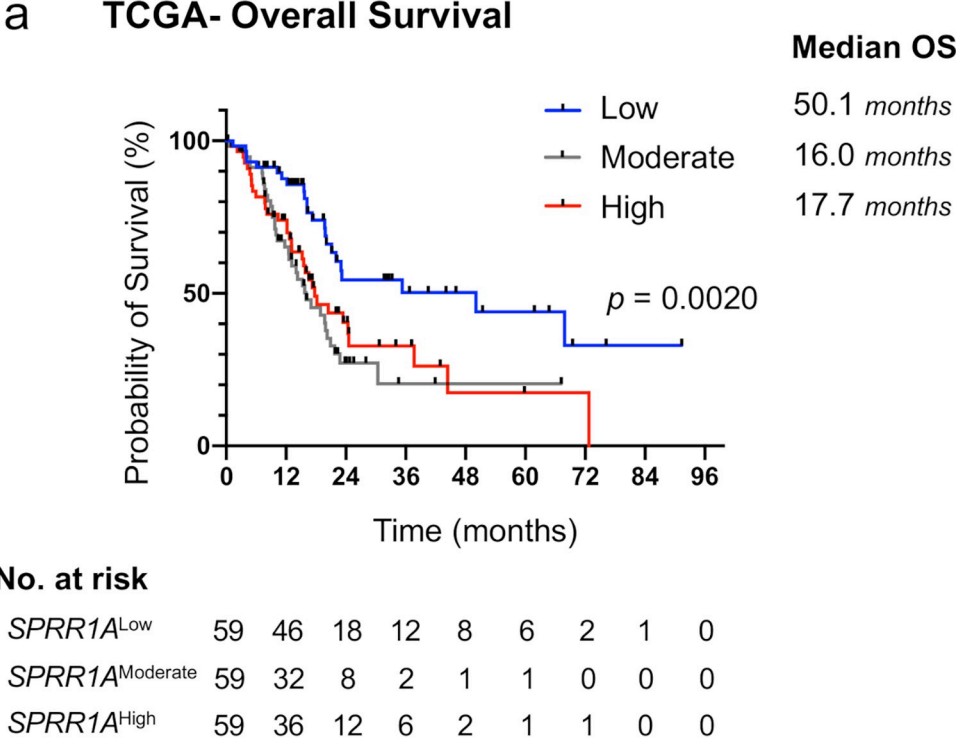

**a** **TCGA- Overall Survival**

**Median OS**

| | |
|---|---|
| Low | 50.1 *months* |
| Moderate | 16.0 *months* |
| High | 17.7 *months* |

*p* = 0.0020

**No. at risk**

| | | | | | | | | | |
|---|---|---|---|---|---|---|---|---|---|
| *SPRR1A*Low | 59 | 46 | 18 | 12 | 8 | 6 | 2 | 1 | 0 |
| *SPRR1A*Moderate | 59 | 32 | 8 | 2 | 1 | 1 | 0 | 0 | 0 |
| *SPRR1A*High | 59 | 36 | 12 | 6 | 2 | 1 | 1 | 0 | 0 |

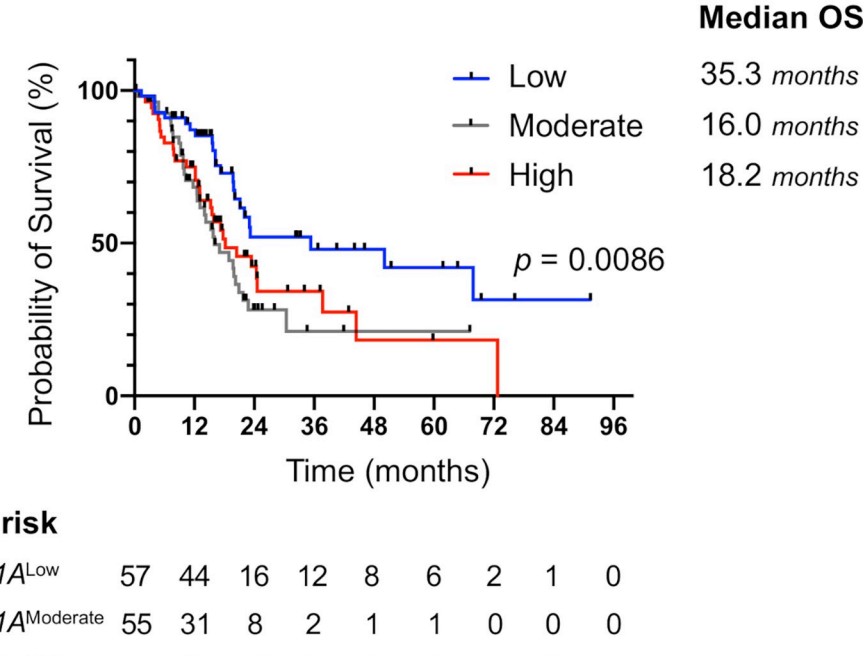

**b** **TCGA- Overall Survival (Except for *KRT5*High cases)**

**Median OS**

| | |
|---|---|
| Low | 35.3 *months* |
| Moderate | 16.0 *months* |
| High | 18.2 *months* |

*p* = 0.0086

**No. at risk**

| | | | | | | | | | |
|---|---|---|---|---|---|---|---|---|---|
| *SPRR1A*Low | 57 | 44 | 16 | 12 | 8 | 6 | 2 | 1 | 0 |
| *SPRR1A*Moderate | 55 | 31 | 8 | 2 | 1 | 1 | 0 | 0 | 0 |
| *SPRR1A*High | 57 | 35 | 12 | 6 | 2 | 1 | 1 | 0 | 0 |

**Fig 3. The analyses of TCGA transcriptome data indicated that an increased expression of SPRR1A predicts a poor prognosis in PDAC patients.** (a) Kaplan-Meier estimates of the OS stratified by the SPRR1A expression in pancreatic cancers. (b) Kaplan-Meier estimates of the OS stratified by the SPRR1A expression in pancreatic cancer patients, except for those with a high KRT5 expression (dotted line in S3C Fig). Tick marks indicate censored data. Kaplan–Meier estimates were compared using a stratified log-rank test.

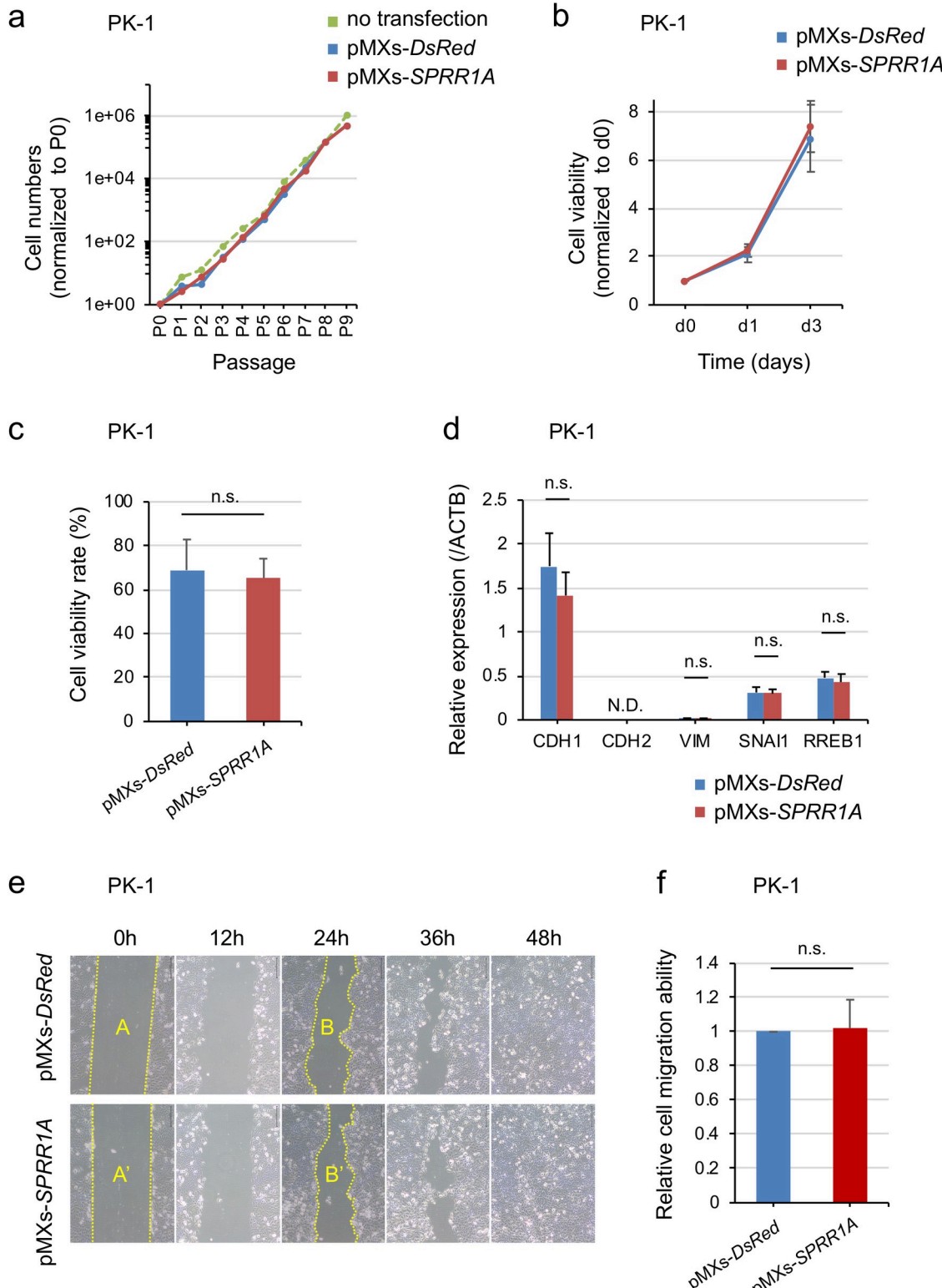

**Fig 4. The overexpression of SPRR1A did not influence cell proliferation, chemo-resistance, EMT or the migration ability in pancreatic cancer cells.** (a) Cell proliferation. The number of SPRR1A-transduced PK-1 cells was counted every 3–4 days after transduction. (b) Cell proliferation assay. The number of viable cells was assessed at days 0, 1 and 3 by measuring cellular ATP levels. (c) Chemo-resistance to Gem. The cell viability in the presence of Gem was calculated as a percentage of the viability in its absence.

(d) The mRNA expression of EMT markers was assessed by quantitative PCR. The expression was normalized to that of β-actin (ACTB). (e) Representative images of wound healing assay at 0, 12, 24, 36 and 48 h. (f) The effects of SPRR1A overexpression on the migration ability were determined by a wound-healing assay. The migration area (MA) in each group was calculated using the Image J software program, according to the following equation: MA = the area of the scratch at 0 h (A, A')–the area of the scratch at 24 h (B, B'). The MA value of the pMXs-*DsRed* population was used as a reference. The following equation determined the relative cell migration ability: Relative cell migration ability = MA (pMXs-*SPRR1A*) / MA (pMXs-*DsRed*). N.D., not detected; n.s., not significant.

and generated a stable line by overexpressing SPRR1A using retroviral vectors. We confirmed that the mRNA and protein levels of SPRR1A were elevated in SPRR1A-transduced cells based on semi-quantitative RT-PCR and Western blotting, respectively (S4A and S4B Fig).

Next, we assessed the changes in the phenotype associated with aggressive behavior, such as cell proliferation, chemo-resistance, epithelial-mesenchymal transition and migration, of the generated cells in vitro. The number of SPRR1A-transduced cells was similar to that of DsRed- or non-transduced cells across passages (Fig 4A). Likewise, cell proliferation assays also revealed that the proliferation of SPRR1A-transduced cells was similar to that of DsRed-transduced cells at 1 and 3 days after seeding (not significant, n = 3) (Fig 4B). To examine the effect of SPRR1A overexpression on the chemo-resistance to Gem, we assessed the difference of viability of SPRR1A- or DsRed-transduced cells in the presence of Gem. There was no significant difference in cell viability following treatment with Gem (not significant, n = 3) (Fig 4C). Next, we checked the epithelial-mesenchymal transition (EMT) markers by real-time quantitative RT-PCR. In SPRR1A-transduced cells, there was a 20% decrease in E-cadherin (CDH1), but this was not statistically significant. Expressions of the other markers were not significantly changed (Fig 4D). Furthermore, the effects of SPRR1A-transduction on the migration ability were determined by a wound-healing assay. There was no marked difference in the speed of wound healing between SPRR1A- and DsRed-transduced cells (not significant, n = 3) (Fig 4E and 4F).

We speculated that native expression of SPRR1A in PK-1 might result in no apparent phenotypic change. When we examined the expression of SPRR1A in various pancreatic cancer cell lines, including PK-1, some lines (KLM-1, Panc-1 and MIAPaca2) did not express SPRR1A (S4E Fig). Among them, we chose Panc-1 and examined the cell proliferation upon the forced expression of SPRR1A (S4F and S4G Fig). Nevertheless, no significant changes were observed, similar to PK-1 (S4H Fig).

## SPRR1A overexpression did not alter the global gene expression profiles associated with aggressive behavior of PDAC

To identify the molecular changes, which are not apparent in the phenotype, caused by SPRR1A overexpression, we performed mRNA sequencing using stable and transient SPRR1A-overexpressing cells to comprehensively examine the changes in the gene expression profiles.

First, we compared the gene expression between the stable SPRR1A- and DsRed-overexpressing cells and identified 70 upregulated entities and 56 downregulated entities with a more than 2-fold change upon stable SPRR1A overexpression (S5A Fig, left). We also compared the gene expression between the transient SPRR1A- and GFP-overexpressing cells and identified 26 upregulated entities and 96 downregulated entities with more than 2-fold change by transient SPRR1A overexpression (S5A Fig, right). We performed a GO analysis for the 70 and 26 upregulated entities and the 56 and 96 downregulated entities but found no significant GO terms satisfying the corrected p-value cut-off of 0.05.

Next, we drew Venn diagrams of the entities more highly expressed in stable and transient overexpression cells than in controls (S5B Fig, above). Except for *SPRR1A*, only one entity

**Table 2. Univariate and multivariate analyses of the overall survival time in patients with pancreatic cancer.**

| Variables | | Number of patients (%) | | Univariate analyses | | | Multivariate analyses | | |
|---|---|---|---|---|---|---|---|---|---|
| | | | | Hazard Ratio (95% CI) | | P value | Hazard Ratio (95% CI) | | P value |
| Age (years) | | | | | | | | | |
| | < 65 | 25 | (29.8%) | | | | | | |
| | ≥ 65 | 59 | (70.2%) | 1.318 | (0.757–2.295) | 0.330 | | | |
| Gender | | | | | | | | | |
| | Female | 34 | (40.5%) | | | | | | |
| | Male | 50 | (59.5%) | 1.163 | (0.703–1.924) | 0.558 | | | |
| BMI (kg/m$^2$) | | | | | | | | | |
| | < 24 | 18 | (21.4%) | | | | | | |
| | ≥ 24 | 66 | (78.6%) | 0.942 | (0.510–1.740) | 0.848 | | | |
| CEA (ng/ml) | | | | | | | | | |
| | ≤ 5 | 58 | (69.0%) | | | | | | |
| | > 5 | 26 | (31.0%) | 0.701 | (0.401–1.225) | 0.212 | | | |
| CA19-9 (U/ml) | | | | | | | | | |
| | ≤ 37 | 20 | (23.8%) | | | | | | |
| | > 37 | 64 | (76.2%) | 1.614 | (0.889–2.931) | 0.116 | 1.524 | (0.835–2.783) | 0.170 |
| Pathological stage[#] | | | | | | | | | |
| | IIa or IIb | 70 | (83.3%) | | | | | | |
| | III | 14 | (16.7%) | 1.11 | (0.560–2.198) | 0.765 | | | |
| T factor | | | | | | | | | |
| | T1 or T2 | 36 | (42.9%) | | | | | | |
| | T3 or T4 | 48 | (57.1%) | 0.736 | (0.451–1.201) | 0.220 | | | |
| N factor | | | | | | | | | |
| | N0 | 17 | (20.2%) | | | | | | |
| | N1 or N2 | 67 | (79.8%) | 1.442 | (0.751–2.768) | 0.272 | | | |
| Histological grade | | | | | | | | | |
| | Well- or moderately differentiated | 75 | (89.3%) | | | | | | |
| | Poorly differentiated | 9 | (10.7%) | 1.203 | (0.547–2.648) | 0.646 | | | |
| Residual tumor status | | | | | | | | | |
| | R0 | 65 | (77.4%) | | | | | | |
| | R1 | 19 | (22.6%) | 2.498 | (1.414–4.415) | 0.00163 | 2.687 | (1.487–4.855) | 0.00106 |
| Peritoneal lavage cytology | | | | | | | | | |
| | CY0 | 84 | (100.0%) | | | | | | |
| | CY1 | 0 | (0.0%) | NA | | NA | | | |
| Neoadjuvant chemotherapy | | | | | | | | | |
| | Absent | 71 | (84.5%) | | | | | | |
| | Present | 13 | (15.5%) | 1.055 | (0.536–2.076) | 0.877 | | | |
| Adjuvant chemotherapy | | | | | | | | | |
| | Absent | 26 | (31.0%) | | | | | | |
| | Present | 58 | (69.0%) | 0.685 | (0.408–1.149) | 0.152 | 0.594 | (0.348–1.013) | 0.0559 |
| SPRR1A | | | | | | | | | |
| | Low expression | 53 | (63.1%) | | | | | | |
| | High expression | 31 | (36.9%) | 1.716 | (1.031–2.856) | 0.0378 | 1.706 | (1.018–2.862) | 0.0427 |

[#]Pathological stage was classified according to the UICC 8th edition.

BMI, body mass index; CEA, carcinoembryonic antigen; CA19-9, carbohydrate antigen 19–9.

NA, not available.

(*SNORD138*) had elevated levels in common between the stable and transient overexpression cells. We also drew Venn diagrams of the entities whose expression was downregulated in stable and transient overexpression cells compared with controls (S5B Fig, below). Seven entities (*mir-let-7i*, *mir-635*, *LOC100131289*, *SNORA2A*, *SNORA103*, *SNORD4B* and *SNORD84*) were identified as commonly downregulated entities in the stable and transient overexpression cells. However, these entities showed no common characteristics.

We intended to elucidate the function of SPRR1A by overexpressing SPRR1A in *vitro* but did not note any significant molecular changes.

## A high expression of SPRR1A may be a hallmark of a novel molecular subtype of PDAC

To identify the gene expression profiles in PDAC with a high SPRR1A expression, we compared the gene expression profiles between the high- (n = 59) and low- (n = 59) SPRR1A-expression groups utilizing the TCGA-PAAD dataset stratified into 3 groups as in the same method as in the prognostic analyses in Fig 3. We identified 345 entities with an elevated expression (p < 0.05, fold change > 3) and 531 with a reduced expression (p < 0.05, fold change > 5) in the high-SPRR1A-expression group compared with the low-SPRR1A-expression group (S4 Table). We performed a GO analysis for the 345 upregulated entities and found GO terms related to squamous epithelium, such as keratinization (GO:0031424, p = 9.58E-12), cornified envelope (GO:0001533, p = 4.91E-11), and skin development (GO:0043588, p = 6.51E-6) (S6A Fig).

PDAC has been proposed to be classified into two molecular subtypes—the "classical" and "basal-like" subtypes—based on transcriptomic data, and these molecular subtypes have been reported to correlate with the patient prognosis [28]. To clarify the relationship between the SPRR1A expression and these molecular subtypes of PDAC, we next examined the expression of SPRR1A and the signature genes of the "classical" and "basal-like" subtypes described by Moffitt et al. [28] using transcriptome data of our *in vitro* experiments and TCGA-PAAD cases. The analyses of transcriptome data of our *in vitro* study indicated that neither stable nor transient SPRR1A overexpression changes the expression of these signature genes (S5 Table). In the analyses of TCGA cases, we found low to medium positive correlations between the expression of SPRR1A and several signature genes in both the "classical" and "basal-like" subtypes (S6B Fig). In addition, we classified TCGA cases into three clusters, clusters 1 ("classical"), 2 ("classical"), and 3 ("basal-like"), and created the heatmap of the expression of SPRR1A and the signature genes of the molecular subtypes of PDAC. This classification of TCGA cases into two molecular subtypes revealed that both molecular subtypes contained similar proportions of cases with a high SPRR1A expression ("classical" 46/96 cases vs. "basal-like" 13/22 cases, p = 0.479) (S6C Fig) and that there was no significant difference in the expression of SPRR1A between the "classical" and "basal-like" subtypes of PDAC (mean FPKM 31.6 vs. 18.9, p = 0.172) (S6D Fig). These results suggest that the increased expression of SPRR1A, which we showed to be associated with a poor prognosis of PDAC in the current study, was independent of the molecular signature reportedly associated with a poor patient prognosis.

## Discussion

In the current study, we found that SPRR1A expression was increased in approximately 35% of pure PDAC specimens without squamous differentiation and that the increased expression of SPRR1A predicts an unfavorable OS of PDAC patients. The increased expression of SPRR1A was independent of the previously reported molecular signature associated with the patient prognosis [28] and correlated with the expression of squamous epithelium-associated

genes, suggesting that a high expression of SPRR1A may be a hallmark of a novel molecular subtype of PDAC. Regarding other types of non-SCCs, including colorectal, breast cancer and diffuse large B-cell lymphoma, previous studies have reported that the SPRR1A expression was increased in 71.9% (82/114) of colorectal cancer, 53.8% (56/111) of breast cancer and 31.5% (305/967) of diffuse large B-cell lymphoma cases, and that a high expression of SPRR1A correlated with a poor prognosis [19–21]. However, to our knowledge, this report is the first to describe the expression of SPRR1A in PDAC and the relationship between SPRR1A expression and the prognosis in PDAC patients. All previous reports as well as our present findings showed that a high SPRR1A expression is associated with a poor, rather than a good, prognosis in non-SCC. Therefore, SPRR1A might be a general prognostic marker in non-SCC.

SPRR1A was mainly expressed in the invasive area of PDAC. We hypothesized that the increased expression of SPRR1A was involved in the aggressive behavior of PDAC. However, our in vitro study showed that SPRR1A overexpression in both PK-1 (expressing SPRR1A) and Panc-1 (not expressing SPRR1A) did not affect the phenotype, such as cell proliferation, chemo-resistance, EMT and migration ability, all of which are associated with aggressive behavior in cancers. Although no study has focused on the biological functions of SPRR1A in any cancer, previous studies have argued that other *SPRR* family genes, such as *SPRR2B* in gastric cancer and *SPRR3* in breast and colorectal cancers, enhance cancer cell proliferation and were involved in cancer growth signaling, such as the AKT, MAPK and MDM2-p53/p21 signaling pathways in vitro [16–18]. We have not evaluated the function of SPRR1A in other types of cancers, but our data suggested that SPRR1A did not induce the activation of cancer growth signaling, at least not in PDAC. We also revealed that SPRR1A overexpression causes only a minor change in gene expression patterns according to RNA sequencing analysis. We noted that the downregulated genes included *mir-let-7i* and *mir-635*, which have been reported to act as tumor suppressors in some types of cancers [29–31]. However, our data suggested that these genes do not significantly impact the phenotype of PDAC. Taken together, out data suggest that the expression of SPRR1A is a consequence, not a cause, of the aggressive behavior of PDAC.

Our data indicated that SPRR1A, a molecule not expressed in the cell of origin of PDAC, was upregulated in some PDAC specimens, resulting in a poor prognosis. This finding suggested that some signaling pathway that regulates SPRR1A might cause the aggressive behavior of some PDACs. Previous studies have indicated that SPRR1A is regulated by p38 MAPK and c-Jun N-Terminal Kinase (JNK) signaling [32,33]. Although ERK1/2 MAPK and JNK signals are known to be core signaling pathways genetically altered in most PDACs [34–36], there are limited reports on p38 MAPK in PDAC. In a future study, we will clarify the association between the SPRR1A expression and the activity of these signaling pathways in PDAC and its impact on PDAC progression.

In conclusion, an increased expression of SPRR1A is associated with a poor prognosis in PDAC patients and may serve as a novel prognostic marker. However, our in vitro study suggests that the expression of SPRR1A is a consequence, not a cause, of the aggressive behavior of PDAC.

## Supporting information

**S1 Fig. The expression of SPRR1A in PDAC cases with squamous differentiation.** (a) The expression of SPRR1A in the normal pancreatic ductal epithelium. Scale bars, 500 μm. (b) The expression of SPRR1A in the normal esophageal epithelium. Scale bars, 500 μm. (c) The expression of SPRR1A and CK5/6 in two PDAC cases with squamous differentiation (Case 12 and 41). Scale bars, 500 μm. (d) A flowchart of the case selection in the current study. (TIF)

**S2 Fig. The OS and RFS in all PDAC cases, including both low and high expression groups of SPRR1A.** (a) Kaplan-Meier estimates of the OS in all PDAC cases. (b) Kaplan-Meier estimates of the RFS in all PDAC cases. Tick marks indicate censored data. (c) Kaplan-Meier estimates of the OS stratified by the SPRR1A expression in PDAC cases, except for those with stage III disease. (d) Kaplan-Meier estimates of the OS stratified by the SPRR1A expression in PDAC cases, except for those with R1.
(TIF)

**S3 Fig. The expression of SPRR1A and KRT5 and the OS in all cases from the TCGA--PAAD data.** (a) Kaplan-Meier estimates of the OS in all cases from the TCGA-PAAD data. (b) Correlation between the transcript level of SPRR1A and KRT5. Correlation analyses were performed using Pearson's product-moment correlation coefficient. (c) The transcript level of SPRR1A and KRT5. The dotted line shows the cases with a high transcript level of KRT5 (above the average plus two s.d.).
(TIF)

**S4 Fig. The expression of SPRR1A in various pancreatic cancer cell lines.** (a) The mRNA expression of SPRR1A in stable SPRR1A-overexpressing cells derived from PK-1 was examined by semi-quantitative RT-PCR using total SPRR1A primers. (b) The protein expression of SPRR1A in stable SPRR1A-overexpressing cells derived from PK-1 was examined by Western blotting. (c) The mRNA expression of SPRR1A in transient SPRR1A-overexpressing cells derived from PK-1 was examined by semi-quantitative RT-PCR using total SPRR1A primers. (d) The protein expression of SPRR1A in transient SPRR1A-overexpressing cells derived from PK-1 was examined by Western blotting. (e) The mRNA expression of SPRR1A in various pancreatic cancer cell lines and normal pancreas tissue was examined by semi-quantitative RT-PCR using endogenous SPRR1A primers. (f) The mRNA expression of SPRR1A in stable SPRR1A-overexpressing cells derived from Panc-1 was examined by semi-quantitative RT-PCR using total SPRR1A primers. (g) The protein expression of SPRR1A in stable SPRR1A-overexpressing cells derived from Panc-1 was examined by Western blotting. (h) Cell proliferation. The cell number of SPRR1A-transduced Panc-1 was counted every three to four days after transduction. RT- indicates control PCR without reverse transcription.
(TIF)

**S5 Fig. A comparison of the gene expression profiles in control and SPRR1A-overexpressing cells derived from PK-1.** (a) Seventy upregulated and 26 downregulated entities were identified with a more than two-fold change in expression by stable SPRR1A overexpression (left). Fifty-six upregulated and 96 downregulated entities were identified with a more than two-fold change in expression by transient SPRR1A overexpression (right). Magenta dots show entities with more than two-fold changes. The scale is shown in base two logarithm. (b) Venn diagrams of the entities more highly expressed in stable and transient overexpression cells than in controls (above). Venn diagrams of the entities whose expression was downregulated in stable and transient overexpression cells than in controls (below).
(TIF)

**S6 Fig. The association between SPRR1A and the signature genes of the molecular subtypes of PDAC.** (a) The blue bars indicate GO terms (p < 0.001) related to the upregulated entities in the high-SPRR1A-expression group in TCGA-PAAD data. (b) The heatmap indicates the result of correlation analyses between SPRR1A and the signature genes of the molecular subtypes of PDAC in TCGA-PAAD cases. (c) The heatmap indicates the expression of SPRR1A and the signature genes of the molecular subtypes of PDAC, classified by K-means cluster analyses in TCGA-PAAD cases. The circled numbers 1, 2, and 3 indicate clusters,

respectively. Clusters 1 and 2 represent the "classical" subtype, while cluster 3 represents the "basal-like" subtype. Arrows indicate cases with a high SPRR1A expression. (d) The comparison of the SPRR1A expression (FPKM) between the "basal-like" (n = 96) and "classical" (n = 22) subtypes indicated in S6C Fig. n.s., not significant, unpaired *t*-test.
(TIF)

**S1 Table. All data of 86 patients analyzed in the current study.**
(XLSX)

**S2 Table. TCGA-PAAD data analyzed in the current study.**
(XLSX)

**S3 Table. Primer sequences used in RT-PCR.**
(XLSX)

**S4 Table. Ensemble ID List of differentially expressed entities in the high-SPRR1A-expression group compared to the low-SPRR1A-expression group.**
(XLSX)

**S5 Table. The expression of SPRR1A and signature genes of the molecular subtypes of PDAC in our transcriptomic data in the stable and transient SPRR1A-overexpressing cells.**
(XLSX)

**S1 File. The original images of gels and blots for plots in S4A to S4G Fig.**
(ZIP)

## Acknowledgments

The GTEx Project was supported by the Common Fund of the Office of the Director of the National Institutes of Health, and by NCI, NHGRI, NHLBI, NIDA, NIMH, and NINDS. The data used for the analyses described in this manuscript were obtained from the GTEx Portal on 07/14/21. We thank all of the members of our laboratory for their scientific comments and valuable discussion and Y. Matsuoka for the administrative support.

## Author Contributions

**Conceptualization:** Kohei Yamakawa, Takashi Aoi.

**Data curation:** Kohei Yamakawa, Atsuhiro Masuda, Hiroaki Yanagimoto, Hirochika Toyama, Takumi Fukumoto.

**Formal analysis:** Kohei Yamakawa.

**Funding acquisition:** Kohei Yamakawa, Michiyo Koyanagi-Aoi, Takashi Aoi.

**Investigation:** Kohei Yamakawa, Keiichiro Uehara.

**Supervision:** Michiyo Koyanagi-Aoi, Yuzo Kodama, Takashi Aoi.

**Validation:** Takashi Aoi.

**Writing – original draft:** Kohei Yamakawa.

**Writing – review & editing:** Michiyo Koyanagi-Aoi, Keiichiro Uehara, Atsuhiro Masuda, Hiroaki Yanagimoto, Hirochika Toyama, Takumi Fukumoto, Yuzo Kodama, Takashi Aoi.

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
