## [Decision Letter · Decision Letter 0]

20 Jan 2022

PONE-D-21-35289Increased expression of SPRR1A is associated with a poor prognosis in pancreatic ductal adenocarcinomaPLOS ONE

Dear Dr. Takashi Aoi,

Thank you for submitting your manuscript to PLOS ONE. After careful consideration, we feel that it has merit but does not fully meet PLOS ONE’s publication criteria as it currently stands. Therefore, we invite you to submit a revised version of the manuscript that addresses the points raised during the review process.

The authors are required to address carefully the comments raised by Reviewers 1 and 2.

We look forward to receiving your revised manuscript.

Kind regards,

Khushboo Irshad, Ph.D.

Academic Editor

PLOS ONE

Journal Requirements:

“This work was supported by grants from JSPS KAKENHI (18H02796; T.A., 20J12977; K.Y.), Research Center Network for Realization of Regenerative Medicine (16817073) from the Japan Agency for Medical Research and Development, AMED (T.A. and M.K-A.), Akira Sakagami Fund for Research and Education, Kobe University Graduate School of Medicine (T.A. and M.K-A.) and Research Assistance Funds from Shinryokukai General Incorporated Association (T.A.).”

“This work was supported by grants from JSPS KAKENHI (18H02796; T.A., 20J12977; K.Y.), Research Center Network for Realization of Regenerative Medicine (16817073) from the Japan Agency for Medical Research and Development, AMED (T.A. and M.K-A.), Akira Sakagami Fund for Research and Education, Kobe University Graduate School of Medicine (T.A. and M.K-A.) and Research Assistance Funds from Shinryokukai General Incorporated Association (T.A.).

5. PLOS requires an ORCID iD for the corresponding author in Editorial Manager on papers submitted after December 6th, 2016. Please ensure that you have an ORCID iD and that it is validated in Editorial Manager. To do this, go to ‘Update my Information’ (in the upper left-hand corner of the main menu), and click on the Fetch/Validate link next to the ORCID field. This will take you to the ORCID site and allow you to create a new iD or authenticate a pre-existing iD in Editorial Manager. Please see the following video for instructions on linking an ORCID iD to your Editorial Manager account: https://www.youtube.com/watch?v=_xcclfuvtxQ.

7. PLOS ONE now requires that authors provide the original uncropped and unadjusted images underlying all blot or gel results reported in a submission’s figures or Supporting Information files. This policy and the journal’s other requirements for blot/gel reporting and figure preparation are described in detail at https://journals.plos.org/plosone/s/figures#loc-blot-and-gel-reporting-requirements and https://journals.plos.org/plosone/s/figures#loc-preparing-figures-from-image-files. When you submit your revised manuscript, please ensure that your figures adhere fully to these guidelines and provide the original underlying images for all blot or gel data reported in your submission. See the following link for instructions on providing the original image data: https://journals.plos.org/plosone/s/figures#loc-original-images-for-blots-and-gels.

Reviewers' comments:

Reviewer's Responses to Questions

**Comments to the Author**

1. Is the manuscript technically sound, and do the data support the conclusions?

Reviewer #1: Yes

Reviewer #2: Partly

2. Has the statistical analysis been performed appropriately and rigorously? 

Reviewer #1: Yes

Reviewer #2: No

3. Have the authors made all data underlying the findings in their manuscript fully available?

Reviewer #1: Yes

Reviewer #2: Yes

4. Is the manuscript presented in an intelligible fashion and written in standard English?

Reviewer #1: No

Reviewer #2: Yes

5. Review Comments to the Author

Reviewer #1: Yokokawa et al demonstrated that SPRR1A, known as a squamous differentiation marker, was upregulated in approximately one third of non-squamous PDAC tissues, which was significantly associated to poor prognosis of the patients with pancreatic cancer. The authors concluded that the upregulation of SPRR1A does not function as a driving force of oncogenic behavior, but can be used as a potent prognostic marker. Although still missing the biological significance, their findings may provide a new therapeutic target from the perspective of clinical settings. There are, however, some points to be improved for the publication in Plos One journal.

Major concerns：

1) It would be better if the authors can further explore the difference of gene expression profile between low and high SPRR1A expression groups utilizing the TCGA database, such as GO analyses. These may provide molecular subtypes of the high SPRR1A group or the upstream pathways of the SPRR1A upregulation, which can reinforce the clinical significance of SPRR1A in PDAC.

2) In Figure 3 A and B, both line charts appear to be the same. Check the original data and ensure that the figure has been presented correctly.

Minor concerns:

1) The quality of figures is not sufficient to be published, which should be improved to higher resolution.

2) Subscripts should be used in H2O2 and ddH2O.

Reviewer #2: The article entitled "Increased expression of SPRR1A is associated with a poor prognosis in pancreatic ductal adenocarcinoma" by Yamakawa et al. supports the role of SPRR1A overexpression as a poor prognosis factor in PDAC although it seems not to be related to cell proliferation, migration, EMT nor chemoresistance. The article is overall well written; however, some issues have to be amended before considering for publication:

Minor points:

-In abstract, please explain what "pathogenesis" refers to.

-Introduction must include most recent bibliography, please update data and citations.

-Also Introduction is very scarce and could be complemented with the link between prognosis and some hallmarks of PDAC like EMT, chemoresistance and a cold and complex microenvironment.

-Line 88. Include some examples of cancers with non-squamous cell carcinoma that exhibit high expression of SPRR1A.

-In the third paragraph of introduction, new proteins, SPRR3 and SPRR2B, appear without any explanation about their link with SPRR1A. Please include the relationship between all proteins.

-Line 105 & 113 what does "consecutive" mean?

-Name of genes must be written in italics

Major points:

-Please include controls used to set best staining conditions for antibodies. Include also a micrograph of controls that show none crossreactions with secondary antibodies.

-Immunohistochemistry seems very subjective because "high" or "little" expression is not an admissible criteria for a scientific research, please provide an objective immune quantification and use a cut-off point as done in TCGA analyses. Line 341, how much is little expression?

-Please justify why TCGA was analyzed using 3 categories (high, moderate and low) and your cohort of patients using low or high.

-Since stage III patients have tumor cells spread their prognosis is different from stage II patients. These different cohorts may be analyzed separately.

-I strongly recommend to use only early stage PDAC patients since stage III patients were also treated with neoadjuvancy and these drugs could modulate expression levels of SPRR1A.

-Remove R1 patients from analyses since they could interfere with prognosis results of SPRR1A as it could be observed in multivariate analysis.

-CA19-9 expression and adjuvant chemotherapy must be removed from multivariate analysis since they are not statistically significant.

-Does SPRR1A overexpression associated to other characteristic factors of any molecular subtype of pancreatic cancer?

6. PLOS authors have the option to publish the peer review history of their article (what does this mean?). If published, this will include your full peer review and any attached files.

Reviewer #1: No

Reviewer #2: No

---

## [Author Response · Author response to Decision Letter 0]

14 Mar 2022

(Please refer to the attached word file "Response to Reviewers.docx" because it contains some figures and tables.)

Point-by-Point responses to Journal Requirements and the reviewers’ comments:

To Journal Requirements:

Response: We revised our manuscript to meet PLOSONE’s style requirements by referring to templates.

2) Please provide additional details regarding participant consent. In the ethics statement in the Methods and online submission information, please ensure that you have specified (1) whether consent was informed and (2) what type you obtained (for instance, written or verbal, and if verbal, how it was documented and witnessed).

Response: The study was a retrospective study of medical records or archived samples, in which all data were anonymized. The ethics committee in our institute allowed a waiver of prospective informed consent, and this study information was disclosed to the public on our hospital website, providing the eligible patients with an opportunity to opt out.

We have now added details regarding the participant consent in the ethics statement in the Materials and Methods and online submission information as follows:

“Surgical specimens were acquired from all 86 patients with stage II or III PDAC who underwent pancreatectomy between March 2011 and January 2017 at Kobe University Hospital. Clinical information for each patient was obtained from chart review. All data were anonymized and are shown in Supplemental Table 1.

This study was approved by the Institutional Review Board (IRB) for Clinical Research at Kobe University Hospital (approval number: B200179) and performed according to the Declaration of Helsinki principles. The IRB allowed a waiver of prospective informed consent, and this study information was disclosed to the public on our hospital website, providing the eligible patients with an opportunity to opt out.” (Page 5, Line 85-93)

3) Thank you for stating the following in the Acknowledgments Section of your manuscript: Please note that funding information should not appear in the Acknowledgments section or other areas of your manuscript. We will only publish funding information present in the Funding Statement section of the online submission form. Please remove any funding-related text from the manuscript and let us know how you would like to update your Funding Statement. Please include your amended statements within your cover letter; we will change the online submission form on your behalf.

Response: We removed funding-related text from the Acknowledgments Section of our manuscript. The funding information has not changed.

4) In your Data Availability statement, you have not specified where the minimal data set underlying the results described in your manuscript can be found. PLOS defines a study's minimal data set as the underlying data used to reach the conclusions drawn in the manuscript and any additional data required to replicate the reported study findings in their entirety. All PLOS journals require that the minimal data set be made fully available. Upon re-submitting your revised manuscript, please upload your study’s minimal underlying data set as either Supporting Information files or to a stable, public repository and include the relevant URLs, DOIs, or accession numbers within your revised cover letter. Any potentially identifying patient information must be fully anonymized. We will update your Data Availability statement to reflect the information you provide in your cover letter.

Response: The datasets analyzed in the current study are available in the GDC Data Portal (https://portal.gdc.cancer.gov), GTEx Portal (https://gtexportal.org/home/datasets) and GEO [GSE186935] (https://www.ncbi.nlm.nih.gov/geo/). All of the data generated by the analysis are included in this article and its Supporting Information files. We have indicated this in our cover letter.

5) PLOS requires an ORCID iD for the corresponding author in Editorial Manager on papers submitted after December 6th, 2016. Please ensure that you have an ORCID iD and that it is validated in Editorial Manager. To do this, go to ‘Update my Information’ (in the upper left-hand corner of the main menu), and click on the Fetch/Validate link next to the ORCID field. This will take you to the ORCID site and allow you to create a new iD or authenticate a pre-existing iD in Editorial Manager.

Response: We registered the ORCID ID for the corresponding author.

6) We note that you have included the phrase “data not shown” in your manuscript. Unfortunately, this does not meet our data sharing requirements. PLOS does not permit references to inaccessible data. We require that authors provide all relevant data within the paper, Supporting Information files, or in an acceptable, public repository. Please add a citation to support this phrase or upload the data that corresponds with these findings to a stable repository (such as Figshare or Dryad) and provide and URLs, DOIs, or accession numbers that may be used to access these data. Or, if the data are not a core part of the research being presented in your study, we ask that you remove the phrase that refers to these data.

Response: We revised the manuscript to be more detailed so that our findings can be replicated and removed the words "data not shown", as follows: 

“We performed a GO analysis for the 70 and 26 upregulated entities and the 56 and 96 downregulated entities but found no significant GO terms satisfying the corrected p-value cut-off of 0.05.” (Page 26, Line 476-478)

7) PLOS ONE now requires that authors provide the original uncropped and unadjusted images underlying all blot or gel results reported in a submission’s figures or Supporting Information files. When you submit your revised manuscript, please ensure that your figures adhere fully to these guidelines and provide the original underlying images for all blot or gel data reported in your submission. In your cover letter, please note whether your blot/gel image data are in Supporting Information or posted at a public data repository, provide the repository URL if relevant, and provide specific details as to which raw blot/gel images, if any, are not available.

Response: We provided the original underlying images for all blot or gel data reported in our manuscript as Supporting Information files (S1 File).

 

To Reviewer #1:

#1-1) It would be better if the authors can further explore the difference of gene expression profile between low and high-SPRR1A-expression groups utilizing the TCGA database, such as GO analyses. These may provide molecular subtypes of the high SPRR1A group or the upstream pathways of the SPRR1A upregulation, which can reinforce the clinical significance of SPRR1A in PDAC.

Response: As suggested, we first stratified TCGA samples by the transcript level of SPRR1A into three groups: high (FPKM > 8.30, n = 59), moderate (FPKM 0.87 to 8.30, n = 59) and low (FPKM < 0.87, n = 59), in the same method as the prognostic analyses in Fig. 3. We compared gene expression profiles between the high and low groups and identified 345 entities with an elevated expression (p < 0.05, fold change > 3) and 531 with a reduced expression (p < 0.05, fold change > 5) in the high-SPRR1A-expression group compared with the low-SPRR1A-expression group (Supplemental Table 4). We performed a GO analysis for the 345 upregulated entities and found GO terms related to squamous epithelium, such as keratinization (GO:0031424, p = 9.58E-12), cornified envelope (GO:0001533, p = 4.91E-11), and skin development (GO:0043588, p = 6.51E-6). In contrast, a GO analysis for the 531 downregulated entities showed GO terms related to neurons, such as synapse (GO:0045202, p = 2.60E-6) and neuron (GO:0043005, p = 6.94E-5). We have now added these data to S-fig. 6a and the following text to the Materials and Methods, Results and Supporting information section of the revised the manuscript:

“For comparing gene expression profiles between the high- and low-SPRR1A-expression groups, we identified differentially expressed entities using an unpaired t-test (p < 0.05). A gene ontology (GO) analysis of the identified entities was performed using g:Profiler (https://biit.cs.ut.ee/gprofiler/gost) [24] to extract significant GO terms (p < 0.001).” (Page 8, Line 170-173)

“A high expression of SPRR1A may be a hallmark of a novel molecular subtype of PDAC

To identify the gene expression profiles in PDAC with a high SPRR1A expression, we compared the gene expression profiles between the high- (n = 59) and low- (n = 59) SPRR1A-expression groups utilizing the TCGA-PAAD dataset stratified into 3 groups as in the same method as in the prognostic analyses in Fig. 3. We identified 345 entities with an elevated expression (p < 0.05, fold change > 3) and 531 with a reduced expression (p < 0.05, fold change > 5) in the high-SPRR1A-expression group compared with the low-SPRR1A-expression group (Supplemental Table 4). We performed a GO analysis for the 345 upregulated entities and found GO terms related to squamous epithelium, such as keratinization (GO:0031424, p = 9.58E-12), cornified envelope (GO:0001533, p = 4.91E-11), and skin development (GO:0043588, p = 6.51E-6) (S-fig.6a).” (Page 26-27, Line 490-501) 

“S6 Fig. The association between SPRR1A and the signature genes of the molecular subtypes of PDAC. (a) The blue bars indicate GO terms (p < 0.001) related to the upregulated entities in the high-SPRR1A-expression group in TCGA-PAAD data.” (Page 39, Line 769-771)

24. Raudvere U, Kolberg L, Kuzmin I, Arak T, Adler P, Peterson H, et al. g:Profiler: a web server for functional enrichment analysis and conversions of gene lists (2019 update). Nucleic Acids Res. 2019;47(W1):W191-W8. Epub 2019/05/09. doi: 10.1093/nar/gkz369. PubMed PMID: 31066453; PubMed Central PMCID: PMC6602461.

#1-2) In Figure 3 A and B, both line charts appear to be the same. Check the original data and ensure that the figure has been presented correctly.

Response: We reviewed Figs. 3A and 3B and confirmed that they were correct. Fig. 3B is a graph created by excluding the 8 cases with a high KRT5 expression from Fig. 3A. As you pointed out, the difference is small, and the line charts in Figs. 3A and 3B appear the same at first glance. We believe that these data indicate that the cases with a KRT5 high expression, which may be PDAC with squamous differentiation, have a negligible impact on the differing prognosis based on the SPRR1A expression. 

#1-3) The quality of figures is not sufficient to be published, which should be improved to higher resolution.

Response: As you pointed out, the PDF image appears to be a low-resolution image that is insufficient for publication. However, if you click on the link in the upper right corner, you can see a higher-resolution image that satisfies the PLOSONE journal submission guideline.

#1-4) Subscripts should be used in H2O2 and ddH2O.

Response: We revised the words, H2O2 and ddH2O, as follows:

“double-distilled H2O (ddH2O)” (Page 5, Line 102)

“0.3% H2O2/methanol” (Page 5, Line 106)

“ddH2O” (Page 6, Line 114, 116, 126, 128)

 

To Reviewer #2:

#2-1) In abstract, please explain what "pathogenesis" refers to.

Response: We used “pathogenesis” to comprehensively refer to malignant behavior, such as the proliferation and invasion ability of PDAC. As you pointed out, this word was unclear and confusing. Therefore, we changed “pathogenesis” to the more concrete phrase “malignant behavior of PDAC” as follows:

“This study elucidated the expression of SPRR1A in PDAC and its effect on the prognosis and malignant behavior of PDAC.” (Page 2, Line 26-28)

#2-2) Introduction must include most recent bibliography, please update data and citations.

Response: We updated the epidemiological data and added the most recent bibliography as follows:

“Pancreatic cancer is a lethal disease with the poorest prognosis, with a 5-year survival rate of approximately 6%-9% [1, 2], in various cancers. The number of deaths caused by pancreatic cancer more than doubled from 1990 to 2017, with 466,000 deaths reported worldwide in 2020 [3, 4].” (Page 3, Line 49-52)

“Consequently, only a few effective molecular-targeted therapies are clinically available for PDAC [7, 8], and treatment options remain limited.” (Page 3, Line 56-58)

2. Siegel RL, Miller KD, Jemal A. Cancer statistics, 2020. CA Cancer J Clin. 2020;70(1):7-30. Epub 2020/01/09. doi: 10.3322/caac.21590. PubMed PMID: 31912902.

4. Sung H, Ferlay J, Siegel RL, Laversanne M, Soerjomataram I, Jemal A, et al. Global Cancer Statistics 2020: GLOBOCAN Estimates of Incidence and Mortality Worldwide for 36 Cancers in 185 Countries. CA Cancer J Clin. 2021;71(3):209-49. Epub 2021/02/05. doi: 10.3322/caac.21660. PubMed PMID: 33538338.

7. Golan T, Hammel P, Reni M, Van Cutsem E, Macarulla T, Hall MJ, et al. Maintenance Olaparib for Germline BRCA-Mutated Metastatic Pancreatic Cancer. N Engl J Med. 2019;381(4):317-27. Epub 2019/06/04. doi: 10.1056/NEJMoa1903387. PubMed PMID: 31157963; PubMed Central PMCID: PMCPMC6810605.

8. Moore MJ, Goldstein D, Hamm J, Figer A, Hecht JR, Gallinger S, et al. Erlotinib plus gemcitabine compared with gemcitabine alone in patients with advanced pancreatic cancer: a phase III trial of the National Cancer Institute of Canada Clinical Trials Group. J Clin Oncol. 2007;25(15):1960-6. Epub 2007/04/25. doi: 10.1200/JCO.2006.07.9525. PubMed PMID: 17452677.

#2-3) Also Introduction is very scarce and could be complemented with the link between prognosis and some hallmarks of PDAC like EMT, chemoresistance and a cold and complex microenvironment.

Response: We added text concerning some hallmarks of PDAC as follows:

“Pancreatic cancer is characterized by intratumor heterogeneity and a highly desmoplastic and immunosuppressive tumor microenvironment, which leads to resistance to chemotherapy and thus a poor prognosis [5, 6].” (Page 3, Line 52-54)

5. Edwards P, Kang BW, Chau I. Targeting the Stroma in the Management of Pancreatic Cancer. Front Oncol. 2021;11:691185. Epub 2021/08/03. doi: 10.3389/fonc.2021.691185. PubMed PMID: 34336679; PubMed Central PMCID: PMCPMC8316993.

6. Binnewies M, Roberts EW, Kersten K, Chan V, Fearon DF, Merad M, et al. Understanding the tumor immune microenvironment (TIME) for effective therapy. Nat Med. 2018;24(5):541-50. Epub 2018/04/25. doi: 10.1038/s41591-018-0014-x. PubMed PMID: 29686425; PubMed Central PMCID: PMCPMC5998822.

#2-4) Line 88. Include some examples of cancers with non-squamous cell carcinoma that exhibit high expression of SPRR1A.

Response: We revised the text and added some examples of cancers with non-squamous cell carcinoma that exhibit a high expression of SPRR1A, as follows:

“and its increased expression has been reported in some types of non-squamous cell carcinoma (non-SCC), such as colorectal cancer and breast cancer [15].” (Page 3, Line 65-66)

#2-5) In the third paragraph of introduction, new proteins, SPRR3 and SPRR2B, appear without any explanation about their link with SPRR1A. Please include the relationship between all proteins.

Response: We added text concerning SPPR family genes as follows:

“The SPRR gene family consists of 10 members, including SPRR1B, six SPRR2, one SPRR3, and one SPRR4, as well as SPRR1A, and all SPPR genes function as specific cornified envelope precursors [15].” (Page 3, Line 68-70)

15. Carregaro F, Stefanini AC, Henrique T, Tajara EH. Study of small proline-rich proteins (SPRRs) in health and disease: a review of the literature. Arch Dermatol Res. 2013;305(10):857-66. Epub 2013/10/03. doi: 10.1007/s00403-013-1415-9. PubMed PMID: 24085571.

#2-6) Line 105 & 113 what does "consecutive" mean?

Response: In this study, we did not intentionally select 86 patients who were convenient in order to draw any conclusions but rather all 86 patients who underwent pancreatectomy within the defined period. We did this to reduce case selection bias. To clarify this point, we revised the text as follows:

“Surgical specimens were acquired from all 86 patients with stage II or III PDAC who underwent pancreatectomy between March 2011 and January 2017 at Kobe University Hospital.” (Page 5, Line 85-87)

“All specimens were acquired from the 86 total individuals with PDAC, excluding cases without formalin-fixed paraffin-embedded (FFPE) samples, as described above.” (Page 5, Line 96-97)

#2-7) Name of genes must be written in italics

Response: We corrected the gene names to italics to distinguish them from proteins and have indicated them in red.

#2-8) Please include controls used to set best staining conditions for antibodies. Include also a micrograph of controls that show none crossreactions with secondary antibodies.

Response: We used normal rabbit IgG as an isotype control for SPRR1A staining to optimize the antibody staining conditions. We have now included a picture of the negative control for SPRR1A in S-Fig.1b and added the relevant information to the revised manuscript, as follows:

“Slides were washed 3 times with 1x PBS, incubated overnight at 4 °C with SPRR1A rabbit antibody (1:200; Abcam plc, Cambridge, UK; catalog number: ab125374) or normal rabbit IgG (FUJIFILM Wako Pure Chemical Corporation, Osaka, Japan; catalog number: 148-09551).” (Page 6, Line 108-110)

“The normal esophageal epithelium was used as a positive control for SPRR1A staining, and isotype IgG was used as a negative control to optimize the antibody staining conditions (S-Fig. 1b).” (Page 7, Line 140-142)

#2-9) Immunohistochemistry seems very subjective because "high" or "little" expression is not an admissible criteria for a scientific research, please provide an objective immune quantification and use a cut-off point as done in TCGA analyses. Line 341, how much is little expression?

Response: The proportion of SPRR1A-positive cells was low in all PDAC species and did not differ markedly. Therefore, we defined “high” and “low” SPRR1A expression by the staining intensity alone. We used the staining intensity of the normal pancreatic ductal epithelium as the cut-off point for “high” and “low.” PDAC specimens with a higher staining intensity of SPRR1A than that of the normal pancreatic ductal epithelium were classified as having a “high” SPRR1A expression, whereas specimens with a staining intensity of SPRR1A equal to or lower than that of the normal pancreatic ductal epithelium were classified as having a “low” SPRR1A expression.

To clarify this point, we revised the sentence as follows:

“PDAC specimens with a higher staining intensity of SPRR1A than the normal pancreatic ductal epithelium were defined as having a high SPRR1A expression, whereas specimens with a staining intensity of SPRR1A equal to or lower than that of the normal pancreatic ductal epithelium were defined as having a low SPRR1A expression.” (Page 7, Line 142-146)

“IHC staining showed that the PDAC regions of 31 (36.9%) specimens, including Cases 3 and 6 (Fig. 1b), were strongly stained for SPRR1A compared to the normal pancreatic ductal epithelium (S-Fig. 1a). In contrast, 53 (63.1%) specimens, including Case 23 (Fig. 1c), exhibited staining equal to or weaker than the normal pancreatic ductal epithelium; we classified the former as the high-SPRR1A-expression group and the latter as the low-SPRR1A-expression group and then used them for the subsequent analyses (detailed in Materials and Methods) (S-fig. 1d).” (Page 15, Line 327-333)

#2-10) Please justify why TCGA was analyzed using 3 categories (high, moderate and low) and your cohort of patients using low or high.

Response: Our cohort evaluation determined the protein expression, whereas a TCGA analysis was used to evaluate the mRNA expression. Obtaining a perfect match between the protein and mRNA expression is difficult. We created a histogram of the cases in the TCGA dataset with SPRR1A expression and noted that the subjects were divided into three groups: cases with little or no expression (FPKM < 1, n = 63), cases with high expression (FPKM > 9, n = 56), and a few cases with middling expression (FPKM 1 to 9, n = 58). Based on this expression distribution, we considered it reasonable to divide the patients into three groups rather than two (see the figure below). Therefore, we performed prognostic analyses using three groups of TCGA samples stratified by the transcript level of SPRR1A. We also tried performing prognostic analyses using TCGA samples stratified into two groups, but the difference in the OS was less clear than when using TCGA samples stratified into three groups (see the figure below). This result also indicated that dividing the patients into three groups was reasonable.

 　 

#2-11) Since stage III patients have tumor cells spread their prognosis is different from stage II patients. These different cohorts may be analyzed separately.

#2-12) I strongly recommend to use only early stage PDAC patients since stage III patients were also treated with neoadjuvancy and these drugs could modulate expression levels of SPRR1A.

Response: As suggested, we excluded stage III patients and reanalyzed our data. In the analysis excluding stage III cases, the OS was significantly lower in the high-SPRR1A-expression group than in the low-SPRR1A-expression group. We revised the sentence as follows and added S-fig. 2c:

“Due to the significant influence of the pathological stage and residual tumor status on the patient prognosis, we excluded stage III and R1 cases, respectively, and assessed the prognostic value of the SPRR1A expression again. In the analysis excluding stage III cases, the OS was significantly lower in the high-SPRR1A-expression group than in the low-SPRR1A-expression group (median OS 22.1 months vs. 33.7 months, p = 0.0322) (S-fig. 2c).” (Page 18, Line 354-358)

“(c) Kaplan-Meier estimates of the OS stratified by the SPRR1A expression in PDAC cases, except for those with stage III disease.” (Page 37-38, Line 734-736)

Following your recommendation, we also performed prognostic analyses using only stage III cases (see the figure below). However, we could not interpret the result due to the fact that there were only a small number of stage III cases, thus making it too small to analyze. Therefore, we did not include these data in the revised manuscript.

#2-13) Remove R1 patients from analyses since they could interfere with prognosis results of SPRR1A as it could be observed in multivariate analysis.

Response: As suggested, we excluded R1 patients and reanalyzed our data. In the analysis excluding R1 cases, the OS was significantly lower in the high-SPRR1A-expression group than in the low-SPRR1A-expression group. We revised the text as below and added S-fig. 2d. These suggestions (stage, R1) improved the quality of our manuscript. However, they did not change our conclusions, as our study found no marked difference in patients' characteristics other than age, and in the univariate and multivariate analyses, SPRR1A was an independent prognostic factor.

“In the analysis excluding R1 cases, the OS was significantly lower in the high-SPRR1A-expression group than in the low-SPRR1A-expression group (median OS 22.0 months vs. 37.0 months, p = 0.0279) (S-fig. 2d).” (Page 18-19, Line 359-361)

“(d) Kaplan-Meier estimates of the OS stratified by the SPRR1A expression in PDAC cases, except for those with R1.” (Page 38, Line 736-737)

#2-14) CA19-9 expression and adjuvant chemotherapy must be removed from multivariate analysis since they are not statistically significant.

Response: A factor can be a confounder even if it is not statistically significant, as it alters the effect of the exposure of interest when included in the model or because it is a confounder only when included with other covariates. A study reported that selecting factors for inclusion in a multivariable model only if the factors are “statistically significant” was not optimal [*1]. Because CA19-9 and adjuvant therapy are well-known prognostic factors [*2, *3], we believe that multivariate analyses including these factors are reasonable. As suggested, we performed a multivariate analysis excluding CA19-9 and adjuvant chemotherapy again and confirmed that this result did not change our conclusion (see the table below). Therefore, we did not include these data in the revised manuscript.

*1 Sun GW, Shook TL, Kay GL. Inappropriate use of bivariable analysis to screen risk factors for use in multivariable analysis. J Clin Epidemiol. 1996;49(8):907-16. Epub 1996/08/01. doi: 10.1016/0895-4356(96)00025-x. PubMed PMID: 8699212.

*2 Ushida Y, Inoue Y, Ito H, Oba A, Mise Y, Ono Y, et al. High CA19-9 level in resectable pancreatic cancer is a potential indication of neoadjuvant treatment. Pancreatology. 2021;21(1):130-7. Epub 2020/12/12. doi: 10.1016/j.pan.2020.11.026. PubMed PMID: 33303373.

*3 Oettle H, Post S, Neuhaus P, Gellert K, Langrehr J, Ridwelski K, et al. Adjuvant chemotherapy with gemcitabine vs observation in patients undergoing curative-intent resection of pancreatic cancer: a randomized controlled trial. JAMA. 2007;297(3):267-77. Epub 2007/01/18. doi: 10.1001/jama.297.3.267. PubMed PMID: 17227978.

#2-15) Does SPRR1A overexpression associated to other characteristic factors of any molecular subtype of pancreatic cancer?

Response: Two molecular subtypes of PDAC, the “classical” and “basal-like” subtype, have been proposed based on transcriptomic data [28]. To clarify the relationship between SPRR1A expression and these molecular subtypes of PDAC, we examined the expression of SPRR1A and the signature genes of these molecular subtypes of PDAC described by Moffitt et al. [28] utilizing RNA sequencing data from our in vitro experiments and TCGA-PAAD cases. The analyses of transcriptome data of our in vitro study indicated that neither stable nor transient SPRR1A overexpression changes the expression of these signature genes (see the figure below and Supplemental Table 5). In the analyses of TCGA cases, we found low to medium positive correlations between the expression of SPRR1A and several signature genes in both the “classical” and “basal-like” subtypes (S-Fig. 6b). In addition, we classified TCGA cases into three clusters, clusters 1 (“classical”), 2 (“classical”), and 3 (“basal-like”), and created the heatmap of the expression of SPRR1A and the signature genes of the molecular subtypes of PDAC. This classification of TCGA cases into two molecular subtypes revealed that both molecular subtypes contained similar proportions of cases with a high SPRR1A expression (“classical” 46/96 cases vs. “basal-like” 13/22 cases, p = 0.479) (S-Fig. 6c) and that there was no significant difference in the expression of SPRR1A between the “classical” and “basal-like” subtypes of PDAC (mean FPKM 31.6 vs. 18.9, p = 0.172) (S-Fig. 6d). These results suggest that the increased expression of SPRR1A, which we showed to be associated with a poor prognosis of PDAC in the current study, was independent of the molecular signature reportedly associated with a poor patient prognosis. We have now added the results of these analyses as S-fig. 6b-d and the following text to the Materials and Methods, Results, Discussion and Supporting information section of the revised the manuscript:

“K-means cluster analyses were carried out using Python 3.7.12 based on the expression of the signature genes of the molecular subtypes of PDAC.” (Page 8, Line 173-175)

“Pearson’s correlation analysis was used to explore the correlation between SPRR1A and the signature genes of the molecular subtypes of PDAC.” (Page 13, Line 290-291)

“PDAC has been proposed to be classified into two molecular subtypes—the “classical” and “basal-like” subtypes—based on transcriptomic data, and these molecular subtypes have been reported to correlate with the patient prognosis [28]. To clarify the relationship between the SPRR1A expression and these molecular subtypes of PDAC, we next examined the expression of SPRR1A and the signature genes of the “classical” and “basal-like” subtypes described by Moffitt et al. [28] using transcriptome data of our in vitro experiments and TCGA-PAAD cases. The analyses of transcriptome data of our in vitro study indicated that neither stable nor transient SPRR1A overexpression changes the expression of these signature genes (Supplemental Table 5). In the analyses of TCGA cases, we found low to medium positive correlations between the expression of SPRR1A and several signature genes in both the “classical” and “basal-like” subtypes (S-Fig. 6b). In addition, we classified TCGA cases into three clusters, clusters 1 (“classical”), 2 (“classical”), and 3 (“basal-like”), and created the heatmap of the expression of SPRR1A and the signature genes of the molecular subtypes of PDAC. This classification of TCGA cases into two molecular subtypes revealed that both molecular subtypes contained similar proportions of cases with a high SPRR1A expression (“classical” 46/96 cases vs. “basal-like” 13/22 cases, p = 0.479) (S-Fig. 6c) and that there was no significant difference in the expression of SPRR1A between the “classical” and “basal-like” subtypes of PDAC (mean FPKM 31.6 vs. 18.9, p = 0.172) (S-Fig. 6d). These results suggest that the increased expression of SPRR1A, which we showed to be associated with a poor prognosis of PDAC in the current study, was independent of the molecular signature reportedly associated with a poor patient prognosis.” (Page 27, Line 502-522)

“The increased expression of SPRR1A was independent of the previously reported molecular signature associated with the patient prognosis [28] and correlated with the expression of squamous epithelium-associated genes, suggesting that a high expression of SPRR1A may be a hallmark of a novel molecular subtype of PDAC.” (Page 29, Line 527-531)

“(b) The heatmap indicates the result of correlation analyses between SPRR1A and the signature genes of the molecular subtypes of PDAC in TCGA-PAAD cases. (c) The heatmap indicates the expression of SPRR1A and the signature genes of the molecular subtypes of PDAC, classified by K-means cluster analyses in TCGA-PAAD cases. The circled numbers 1, 2, and 3 indicate clusters, respectively. Clusters 1 and 2 represent the "classical" subtype, while cluster 3 represents the "basal-like" subtype. Arrows indicate cases with a high SPRR1A expression. (d) The comparison of the SPRR1A expression (FPKM) between the "basal-like" (n = 96) and "classical" (n = 22) subtypes indicated in S-fig. 6c. n.s., not significant, unpaired t-test.” (Page 39, Line 771-779)

28. Moffitt RA, Marayati R, Flate EL, Volmar KE, Loeza SG, Hoadley KA, et al. Virtual microdissection identifies distinct tumor- and stroma-specific subtypes of pancreatic ductal adenocarcinoma. Nat Genet. 2015;47(10):1168-78. Epub 2015/09/08. doi: 10.1038/ng.3398. PubMed PMID: 26343385; PubMed Central PMCID: PMCPMC4912058.

---

## [Decision Letter · Decision Letter 1]

24 Mar 2022

Increased expression of SPRR1A is associated with a poor prognosis in pancreatic ductal adenocarcinoma

PONE-D-21-35289R1

Dear Dr. Takashi Avi,

We’re pleased to inform you that your manuscript has been judged scientifically suitable for publication and will be formally accepted for publication once it meets all outstanding technical requirements.

Kind regards,

Khushboo Irshad, Ph.D.

Academic Editor

PLOS ONE

Additional Editor Comments (optional):

The authors have appropriately addressed the comments raised by both the authors.

Reviewers' comments:

Reviewer's Responses to Questions

**Comments to the Author**

1. If the authors have adequately addressed your comments raised in a previous round of review and you feel that this manuscript is now acceptable for publication, you may indicate that here to bypass the “Comments to the Author” section, enter your conflict of interest statement in the “Confidential to Editor” section, and submit your "Accept" recommendation.

Reviewer #1: All comments have been addressed

2. Is the manuscript technically sound, and do the data support the conclusions?

Reviewer #1: Yes

3. Has the statistical analysis been performed appropriately and rigorously? 

Reviewer #1: N/A

4. Have the authors made all data underlying the findings in their manuscript fully available?

Reviewer #1: Yes

5. Is the manuscript presented in an intelligible fashion and written in standard English?

Reviewer #1: Yes

6. Review Comments to the Author

Reviewer #1: The authors have substantially revised the manuscript in response to reviewer comments, and this version is significantly improved for the publication.

7. PLOS authors have the option to publish the peer review history of their article (what does this mean?). If published, this will include your full peer review and any attached files.

Reviewer #1: No

---

## [Editor Report · Acceptance letter]

19 May 2022

PONE-D-21-35289R1 

Increased expression of SPRR1A is associated with a poor prognosis in pancreatic ductal adenocarcinoma 

Dear Dr. Aoi:

I'm pleased to inform you that your manuscript has been deemed suitable for publication in PLOS ONE. Congratulations! Your manuscript is now with our production department. 

Kind regards, 

on behalf of

Dr. Khushboo Irshad 

Academic Editor

PLOS ONE